# Enhancing the consistency of spaceborne and ground-based radar comparisons by using beam blockage fraction as a quality filter

Irene Crisologo[1], Robert A. Warren[2], Kai Mühlbauer[3], and Maik Heistermann[1]

[1]Institute of Earth and Environmental Sciences, University of Potsdam, Germany
[2]School of Earth, Atmosphere and Environment, Monash University, Australia
[3]Meteorological Institute, University of Bonn, Germany

**Correspondence:** Irene Crisologo (crisologo@uni-potsdam.de)

**Abstract.**

We explore the potential of spaceborne radar (SR) observations from the Ku-band precipitation radars on board the TRMM and GPM satellites as a reference to quantify the ground radar (GR) reflectivity bias. To this end, the 3D volume-matching algorithm proposed by Schwaller and Morris (2011) is implemented and applied to five years (2012-2016) of observations. We further extend the procedure by a framework to take into account the data quality of each ground radar bin. Through these methods, we are able to assign a quality index to each matching SR–GR volume, and thus compute the GR calibration bias as a quality-weighted average of reflectivity differences in any sample of matching GR–SR volumes. We exemplify the idea of quality-weighted averaging by using the beam blockage fraction as the basis of a quality index. As a result, we can increase the consistency of SR and GR observations, and thus the precision of calibration bias estimates. The remaining scatter between GR and SR reflectivity, as well as the variability of bias estimates between overpass events indicate, however, that other error sources are not yet fully addressed. Still, our study provides a framework to introduce any other quality variables that are considered relevant in a specific context. The code that implements our analysis is based on the open source software library *wradlib*, and is, together with the data, publicly available to monitor radar calibration, or to scrutinize long series of archived radar data back to December 1997, when TRMM became operational.

## 1 Introduction

Weather radars are essential tools in providing high quality information about precipitation with high spatial and temporal resolution in three dimensions. However, several uncertainties deteriorate the accuracy of rainfall products, with calibration contributing the most amount (Houze Jr et al., 2004), while also varying in time (Wang and Wolff, 2009). While adjusting ground radars (GR) by comparison with a network of rain gauges (also know as *gauge adjustment*) is a widely used method, it suffers from representativeness issues. Furthermore, gauge-adjustment accumulates uncertainties along the entire rainfall estimation chain (e.g. including the uncertain transformation from reflectivity to rainfall rate), and thus does not provide a direct

reference for the measurement of reflectivity. Relative calibration (defined as the assessment of bias between the reflectivity of two radars) has been steadily gaining popularity, in particular the comparison with space-borne precipitation radars (SR) (such as the precipitation radar on-board the Tropical Rainfall Measuring Mission (TRMM; 1997–2014; Kummerow et al. (1998)) and the dual-frequency Precipitation Radar on the subsequent Global Precipitation Measurement mission (GPM; 2014–present; Hou et al. (2013))). Several studies have shown that surface precipitation estimates from GRs can be reliably compared to precipitation estimates from SRs for both TRMM (Amitai et al., 2009; Joss et al., 2006; Kirstetter et al., 2012) and GPM (Gabella et al., 2017; Petracca et al., 2018; Speirs et al., 2017). In addition, a major advantage of relative calibration and gauge adjustment in contrast to the absolute calibration (i.e. minimizing the bias in measured power between an external or internal reference noise source and the radar at hand) is that they can be carried out a posteriori, and thus be applied to historical data.

Since both ground radars and space-borne precipitation radars provide a volume-integrated measurement of reflectivity, a direct comparison of the observations can be done in three dimensions (Anagnostou et al., 2001; Gabella et al., 2006, 2011; Keenan et al., 2003; Warren et al., 2018). Moreover, as the spaceborne radars are and have been constantly monitored and validated (with their calibration accuracy proven to be consistently within 1 dB) (TRMM: Kawanishi et al. (2000); Takahashi et al. (2003); GPM: Furukawa et al. (2015); Kubota et al. (2014); Toyoshima et al. (2015)), they have been suggested as a suitable reference relative calibration of ground radars (Anagnostou et al., 2001; Islam et al., 2012; Liao et al., 2001; Schumacher and Houze Jr, 2003).

Relative calibration between SRs and GRs was originally suggested by Schumacher and Houze (2000), but the first method to match SR and GR reflectivity measurements was developed by Anagnostou et al. (2001). In their method, SR and GR measurements are resampled to a common three-dimensional grid. Liao et al. (2001) developed a similar resampling method. Such 3D-resampling methods have been used in comparing SR and GR for both SR validation and GR bias determination (Bringi et al., 2012; Gabella et al., 2006, 2011; Park et al., 2015; Wang and Wolff, 2009; Zhang et al., 2018; Zhong et al., 2017) Another method was suggested by Bolen and Chandrasekar (2003) and later on further developed by Schwaller and Morris (2011), where the SR–GR matching is based on the geometric intersection of SR and GR beams. This geometry matching algorithm confines the comparison to those locations where both instruments have actual observations, without interpolation or extrapolation. The method has also been used in a number of studies comparing SR and GR reflectivities (Chandrasekar et al., 2003; Chen and Chandrasekar, 2016; Islam et al., 2012; Kim et al., 2014; Wen et al., 2011). A sensitivity study by Morris and Schwaller (2011) found that method to give more precise estimates of relative calibration bias as compared to grid-based methods.

Due to different viewing geometries, ground radars and spaceborne radars are affected by different sources of uncertainty and error. Observational errors with regard to atmospheric properties such as reflectivity are, for example, caused by ground clutter or partial beam blocking. Persistent systematic errors in the observation of reflectivity by ground radars are particularly problematic: the intrinsic assumption of the bias estimation is that the only systematic source of error is radar calibration. It is therefore particularly important to address such systematic observation errors.

In this study, we demonstrate that requirement with the example of partial beam blocking. The analysis is entirely based on algorithms implemented in the open source software library *wradlib* (Heistermann et al., 2013b), including a technique to infer

partial beam blocking by simulating the interference of the radar beam with terrain surface based on a digital elevation model.

Together, that approach might become a reference for weather services around the world who are struggling to create unbiased radar observations from many years of archived single-polarized radar data, or to consistently monitor the bias of their radar observations. We demonstrate the approach in a case study with five years of data from the single-polarized S-band radar near the city of Subic, Philippines, which had been shown in previous studies to suffer from substantial miscalibration (Abon et al., 2016; Heistermann et al., 2013a).

## 2 Data

### 2.1 Space-Borne Precipitation Radar

Precipitation radar data were gathered from TRMM 2A23 and 2A25 version 7 products (NASA, 2017) for overpass events intersecting with the Subic ground radar coverage between 1 June 2012 to 30 September 2014, and GPM 2AKu version 5A products (Iguchi et al., 2010) from 1 June 2014 to 31 December 2016. Ka band observations have not been considered due to higher susceptibility to attenuation, and a limited validity of Rayleigh scattering in a substantial portion of rainfall cases (Baldini et al., 2012). From the collection of overpasses within these dates, only 183 TRMM overpasses and 103 GPM passes were within the radar coverage. The data were downloaded from NASA's Precipitation Processing System (PPS) through the STORM web interface (https://storm.pps.eosdis.nasa.gov/storm/) on 15 February 2018 for TRMM and 14 June 2018 for GPM. The parameters of TRMM/GPM extracted for the analysis are the same as Warren et al. (2018; their Table 3).

It is important to note that, at the time of writing, changes in calibration parameters applied in the GPM Version 5 products resulted in an increase of +1.1 dB from the corresponding TRMM version 7 products (NASA, 2017).

### 2.2 Ground Radar

The Philippine Atmospheric, Geophysical, and Astronomical Services Administration (PAGASA) maintains a nationwide network of ten weather radars, eight of which are single-polarization S-band radars and two are dual-polarization C-band radars. Subic radar, which covers the greater Metropolitan Manila area, has the most extensive set of archived data. The radar coverage includes areas that receive some of the highest mean annual rainfall in the country.

The Subic radar sits on top of a hill at 532 m.a.s.l. in the municipality of Bataan, near the border to Zambales (location: 14.82 °N, 120.36 °E) (see Figure 1). To its south stands Mt. Natib (1253 m.a.s.l.) and to its north runs the Zambales Mountain Range (highest peak stands at 2037 m.a.s.l.). To the west is the Redondo Peninsula in the southern part of the Zambales province, where some mountains are also situated. Almost half of the coverage of the Subic radar is water, with Manila Bay to its southeast and the West Philippine Sea to the west. Technical specifications of the radar are summarized in Table 1. Data from April 2012 to December 2016 were obtained from PAGASA. Throughout the five years the scan strategy remained the same, except for 2015 when it was limited to only three elevation angles per volume due to hardware issues. The standard scanning strategy was re-implemented in 2016.

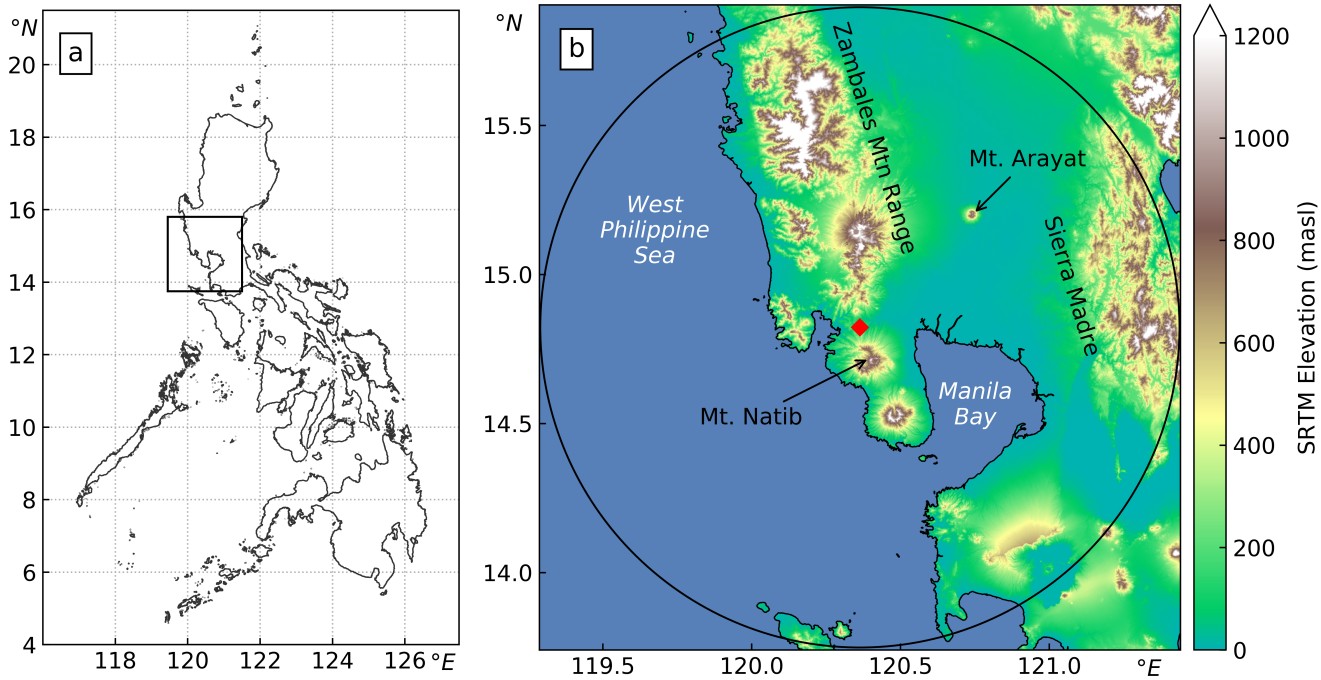

**Figure 1.** (a) Map of the Philippines showing the region of study and (b) the 120km coverage of the Subic radar (location marked with red diamond) with the SRTM Digital Elevation Model of the surrounding area.

## 3 Method

### 3.1 Partial beam shielding and quality index based on beam blockage fraction

In an ideal situation, SR and GR should have the same measurements for the same volume of the atmosphere, as they are measuring the same target. However, observational differences may arise due to different view geometries, different operating frequencies, different environmental conditions of each instrument, and different processes along the propagation path of the beam. As pointed out before, we focus on beam blockage as an index of GR data quality.

In regions of complex topography, ground radars are typically affected by the effects of beam blockage, induced by the interaction of the beam with the terrain surface resulting into a weakening or even loss of the signal. To quantify that process within the Subic radar coverage, a beam blockage map is generated following the algorithm proposed by Bech et al. (2003). It assesses the extent of occultation using a digital elevation model (DEM). While Bech et al. (2003) used the GTOPO30 DEM at a resolution of around on kilometer, higher DEM resolutions are expected to increase the accuracy of estimates of beam blockage fraction, as shown by Kucera et al. (2004), in particular the near range of the radar (Cremonini et al., 2016). The DEM used in this study is from the Shuttle Radar Topography Mission (SRTM) data, with 1-arc-second (approximately 30-meter) resolution. The DEM was resampled to the coordinates of the radar bin centroids, using spline interpolation, in order

**Table 1.** Characteristics of the Subic radar and its volume scan strategy. The numbers in parentheses correspond to scans in 2015, where the scanning strategy was different due to hardware issues.

|  | Subic Radar |
| --- | --- |
| Polarization | Single-Pol |
| Position (lat/lon) | $14.82°N\ 120.36\ °E$ |
| Altitude | 532 m.a.s.l. |
| Maximum Range | 120 km (150 km) |
| Azimuth resolution | 1 ° |
| Beam width | 0.95 ° |
| Gate length | 500 m (250 m) |
| Number of elevation angles | 14 (3) |
| Elevation angles | 0.5, 1.5, 2.4, 3.4, 4.3, 5.3, 6.2, 7.5, 8.7, 10, 12, 14, 16.7, 19.5 (°) |
|  | (0.0, 1.0, 2.0) |
| Volume cycle interval | 9 minutes |
| Data available since | April 2012 |
| Peak power | 850 kW |
| Wavelength | 10.7 cm |

5 to match the polar resolution of the radar data (500 m in range and 1° in azimuth, extending to a maximum range of 120 km from the radar site; see Figure 1). A beam blockage map is generated for all available elevation angles.

The beam blockage fraction was calculated for each bin and each antenna pointing angle. The cumulative beam blockage was then calculated along each ray. A cumulative beam blockage fraction (BBF) of 1.0 corresponds to full occlusion, and a value of 0.0 to perfect visibility.

10 The quality index based on beam blockage fraction is then computed following Zhang et al. (2011) as:

$$Q_{BBF} = \begin{cases} 1 & BBF \leq 0.1 \\ 1 - \frac{BBF - 0.1}{0.4} & 0.1 < BBF \leq 0.5 \\ 0 & BBF > 0.5 \end{cases} \tag{1}$$

A slightly different formulation to transform partial beam blockage to a quality index has been presented in other studies (Figueras i Ventura and Tabary, 2013; Fornasiero et al., 2005; Ośródka et al., 2014; Rinollo et al., 2013) where the quality is zero (0) if BBF is above a certain threshold, and then linearly increases to one (1) above that threshold. It should be noted that these approaches are equally valid and can be used in determining the quality index based on beam blockage.

Figure 2 shows the beam blockage map for the two lowest elevation angles of each scanning strategy. Figure 2a and c are for 0.0° and 1.0°, which are the two lowest elevation angles in 2015, while Figure 2b and d are for 0.5° and 1.5°, which are the two lowest elevation angles for the rest of the dataset.

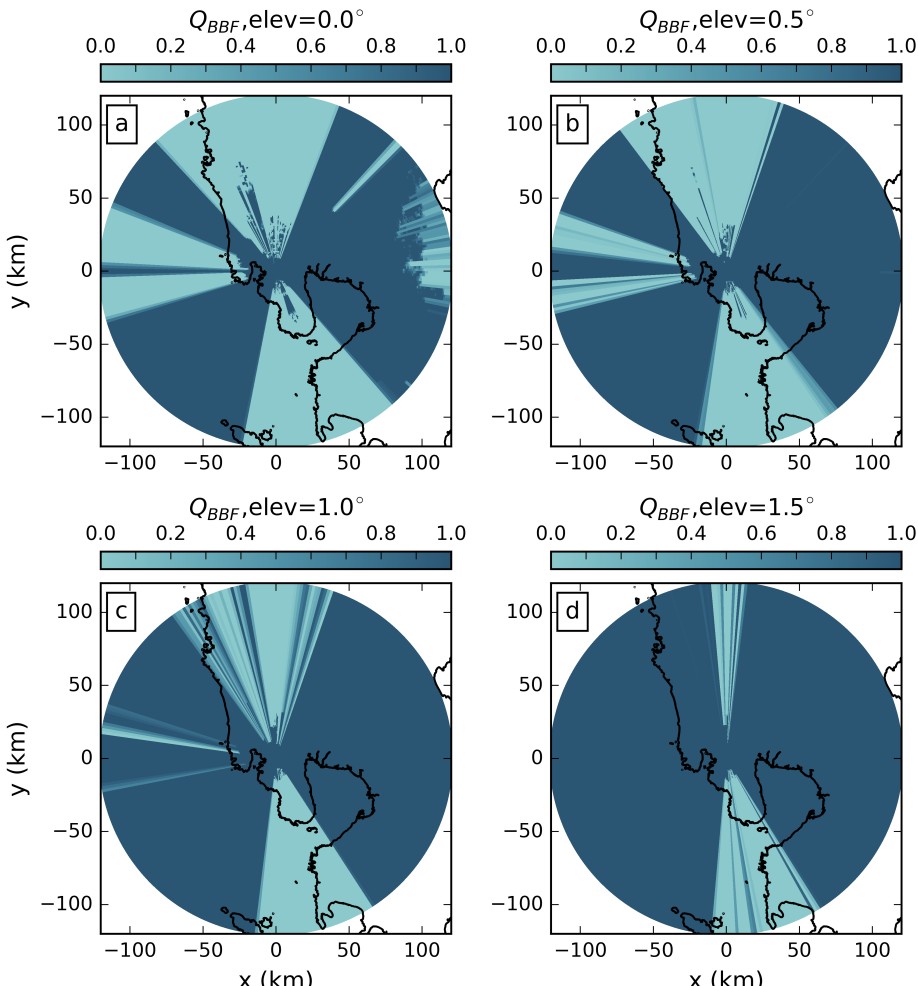

**Figure 2.** Quality index map of the beam blockage fraction for the Subic radar at (a) $0.0°$ (b) $0.5°$ (c) $1.0°$ and (d) $1.5°$ elevation angles.

As expected, the degree of beam blockage decreases with increasing antenna elevation, yielding the most pronounced beam blockage at $0.0°$. Each blocked sector can be explained by the topography (see Figure 1), with the Zambales Mountain Range causing blockage in the northern sector, Mt. Natib in the southern sector, and the Redondo peninsula mountains in the western sector. The Sierra Madre mountains also cause some partial beam blocking at the far east, and a narrow partial blocking northeast of the station where Mount Arayat is located. As the elevation angle increases, the beam blockage becomes less pronounced or even disappears. Substantial blockage persists, however, for the higher elevation angles in the northern and southern sectors.

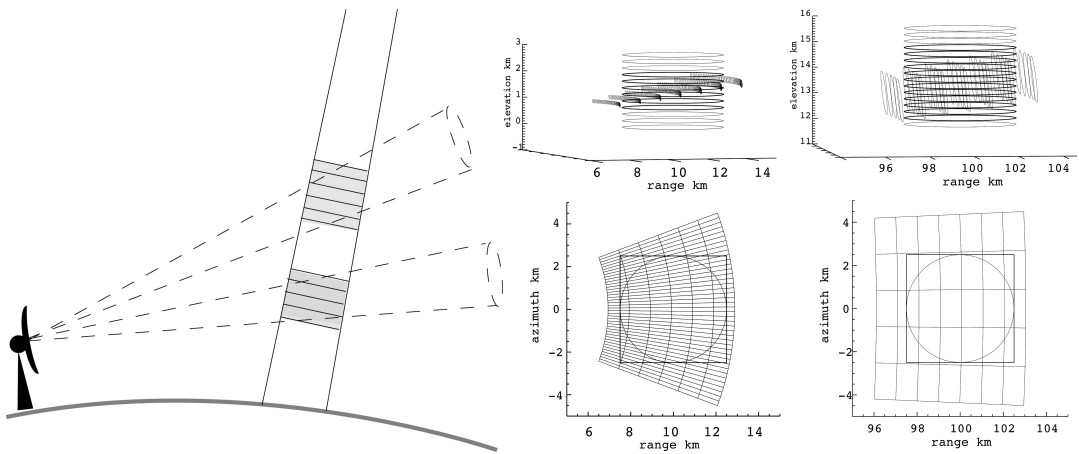

**Figure 3.** Diagram illustrating the geometric intersection. Left panel shows a single SR beam intersecting GR sweeps of two different elevation angles. The two top right panels illustrate the intersection of SR-GR sample volumes in the near and far ranges and the two bottom right panels show the projection of these intersections along an SR ray. From Schwaller and Morris (2011) ©American Meteorological Society. Used with permission.

## 3.2 SR–GR Volume Matching

SR and GR data were matched only for the wet period within each year, which is from June to December. Several meta-data parameters were extracted from the TRMM 2A23 and GPM 2AKu products for each SR gate, such as the corresponding ray's bright band height and width, gate coordinates in three dimensions (longitude and latitude of each ray's Earth intercept and range gate index), time of overpass, precipitation type (*stratiform*, *convective*, or *other*), and rain indicators (*rain certain* or *no-rain*). The parallax-corrected altitude (above mean sea level) and horizontal location (with respect to the GR) of each gate

were determined as outlined in the appendix of Warren et al. (2018). From the bright band height/width and the altitude of each SR gate, the bright band membership of each gate was calculated by grouping all rays in an overpass and computing the mean brightband height and width. A ratio value of less than zero indicates that the gate is below bright band, greater than one indicates that the gate is above the bright band, and a value between zero and one means that the gate is within the bright band. Only gates below and above the brightband were considered in the comparison. Warren et al. (2018) found a positive bias

in GR–SR reflectivity difference for volume-matched samples within the melting layer, compared to those above and below the melting layer. They speculated that this was due to underestimation of the Ku- to S-band frequency correction for melting snow. In addition, while usually the samples above the brightband are used in GPM validation, there are significantly more samples below the melting layer, especially in a tropical environment such as the Philippines. To ensure that there are sufficient bins with actual rain included in the comparison, overpasses with less than 100 gates flagged as rain certain were discarded.

For each SR overpass, the GR sweep with the scan time closest to the overpass time within a 10-min window (±5-min from overpass time) was selected. Both the SR and GR data were then geo-referenced into a common azimuthal equidistant projection centered on the location of the ground radar.

**Table 2.** Filtering criteria for the matching workflow.

| Criteria | Condition |
|---|---|
| Minimum number of pixels in overpass tagged as 'rain' | 100 |
| Bright band membership | below or above |
| GR range limits (min–max) | 15 – 115 km |
| Minimum fraction of bins above minimum SR sensitivity | 0.7 |
| Minimum fraction of bins above minimum GR sensitivity | 0.7 |
| Maximum time difference between SR and GR | 5 min |
| Minimum PR reflectivity | 18 dBZ |

In order to minimize systematic differences in comparing the SR and GR reflectivites caused by the different measuring frequencies, the SR reflectivities were converted from Ku to S Band following the formula:

$$Z(S) = Z(Ku) + \sum_{i=0}^{4} a_i [Z(Ku)]^i \tag{2}$$

where the $a_i$ are the coefficients for dry snow and dry hail, rain, and in between at varying melting stages (Table 1 of Cao et al. (2013)). We used the coefficients for snow in the reflectivity conversion above the brightband, following Warren et al. (2018).

The actual volume matching algorithm closely follows the work of Schwaller and Morris (2011), where SR reflectivity is spatially and temporally matched with GR reflectivity without interpolation. The general concept is highlighted by Figure 3: Each matching sample consists of bins from only *one SR ray* and *one GR sweep*. From the SR ray, those bins were selected that intersect with the vertical extent of a specific GR sweep at the SR ray location. From each GR sweep, those bins were selected that intersect with the horizontal footprint of the SR ray at the corresponding altitude. The SR and GR reflectivity of each matched volume was computed as the average reflectivity of the intersecting SR and GR bins.

The nominal minimum sensitivity of both TRMM PR and GPM KuPR is 18 dBZ, so only values above this level were considered in the calculation of average SR reflectivity in the matched volume. In addition, the fraction of SR gates within a matched volume above that threshold was also recorded. On the other hand, all GR bins are included in the calculation of average GR reflectivity, after setting the bins with reflectivities below 0.0 dBZ to 0.0 dBZ, as suggested by Morris and Schwaller (2011). The filtering criteria applied in the workflow are summarized in Table 2.

### 3.3 Assessment of the average reflectivity bias

Beam blockage and the corresponding GR quality maps were computed for each GR bin (cf. Section 3.1.) For each matched SR–GR volume, the data quality was then based on the minimum quality of the GR bins in that volume.

To analyze the effect of data quality on the estimation of GR calibration bias, we compared two estimation approaches: a simple mean bias that does not take into account beam blockage, and a weighted mean bias that considers the quality value of

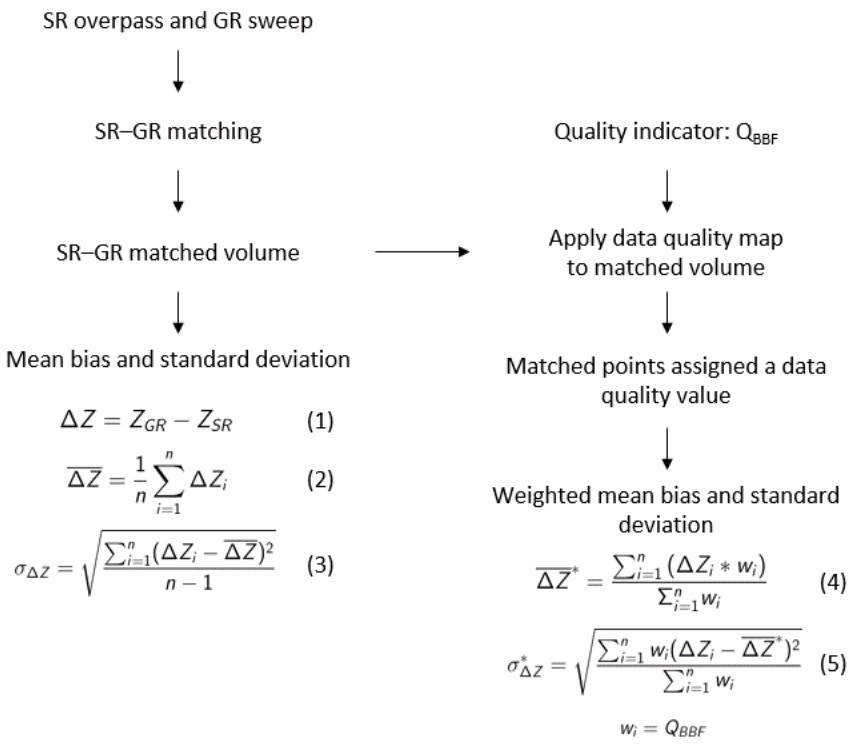

**Figure 4.** Flowchart describing the processing steps to calculate the mean bias and the weighted mean bias between ground radar data and satellite radar data. The results of each step are shown in Section 4.

each sample as weights. The corresponding standard deviation and weighted standard deviation were calculated as well. The overall process is summarized in Figure 4. This way, we provide an overview of the variability of our bias estimates over time.

### 3.4 Computational details

In order to promote transparency and reproducibility of this study, we mostly followed the guidelines provided by Irving (2016) which have also been implemented by a number of recent studies (Blumberg et al., 2017; Irving and Simmonds, 2016; Rasp et al., 2018).

The entire processing workflow is based on *wradlib* (Heistermann et al., 2013b), an extensively documented open-source software library for processing weather radar data. At the time of writing, we used version 1.0.0 released on 01 April 2018, based on Python 3.6. The main dependencies include Numerical Python (NumPy; Oliphant (2015)), Matplotlib (Hunter, 2007), Scientific Python (SciPy; Jones et al. (2014)), h5py (Collette, 2013), netCDF4 (Rew et al., 1989), and gdal (GDAL Development Team, 2017).

5 Reading the TRMM 2A23 and 2A25 version 7 data, GPM 2AKu version 5A data, and the Subic ground radar data in the netCDF format converted through the EDGE software of EEC radars was done through the input/output module of wradlib. The beam blockage modeling is based on the Bech et al. (2003) method implemented as a function in wradlib's data quality module. The volume-matching procedure is built upon the georeferencing and zonal statistics modules, accompanied by Pandas (McKinney, 2010) for organizing and analyzing the resulting database of matched bins. Visualization was carried out with the 10 help of matplotlib (Hunter, 2007) and Py-ART (Helmus and Collis, 2016).

An accompanying GitHub repository that hosts the Jupyter notebooks of the workflow and sample data is made available at `https://github.com/wradlib/radargpm-beamblockage`.

## 4 Results and Discussion

### 4.1 Single event comparison

15 From the 183 TRMM and 103 GPM overpasses that intersected with the 120 km Subic radar range, only 74 TRMM and 40 GPM overpasses were considered valid after applying the selection criteria listed in Table 2. In order to get a better idea about the overall workflow, we first exemplify the results for two specific overpass events—one for TRMM, and one for GPM.

#### 4.1.1 Case 1: 08 November 2013

For the TRMM overpass event on November 8, 2013, the top row of Figure 5 shows SR (a) and GR (b) reflectivity as well as 20 the resulting differences (c) for matching samples at an elevation angle of 0.5°. Each circle in the plots represents a matched volume. A corresponding map of $Q_{BBF}$ is shown in (d) while (e) shows a scatter plot of GR versus SR reflectivities, with points coloured according to their $Q_{BBF}$. The reflectivity difference map and scatter plot indicate significant variability with absolute differences of up to and exceeding 10 dB. Large differences can be observed at the edges of the southern sector affected by beam blockage (cf. also Figure 2). Major parts of that sector did not receive any signal due to total beam blockage, 25 highlighted in Figure 5a with black circles showing the bins where the GR did not obtain valid observations. At the edges, however, partial beam blockage caused substantially lower GR reflectivity values. As expected, large negative differences of $Z_{GR}-Z_{SR}$ are characterized by low quality.

Consequently, the estimate of the calibration bias substantially depends on the consideration of partial beam blockage (or quality). Ignoring quality (simple mean) yields a bias estimate of -1.9 dB while the quality-weighted average yields a bias 30 estimate of -1.2 dB. Accordingly, the standard deviation is reduced from 3.4 to 2.6 dB, indicating a more precise bias estimate.

This case demonstrates how partial beam blockage affects the estimation of GR calibration bias. At a low elevation angle, substantial parts of the sweep are affected by *total* beam blockage. The affected bins are either below the detection limit, or they do not exceed the GR threshold specified in Table 2. As a consequence, these bins will not be considered in the matched samples and will thus not influence the bias estimate, irrespective of using partial beam blockage as a quality filter. At a higher elevation angle, though, the same bins might not be affected by *total* beam blockage, but by *partial* beam blockage, as also

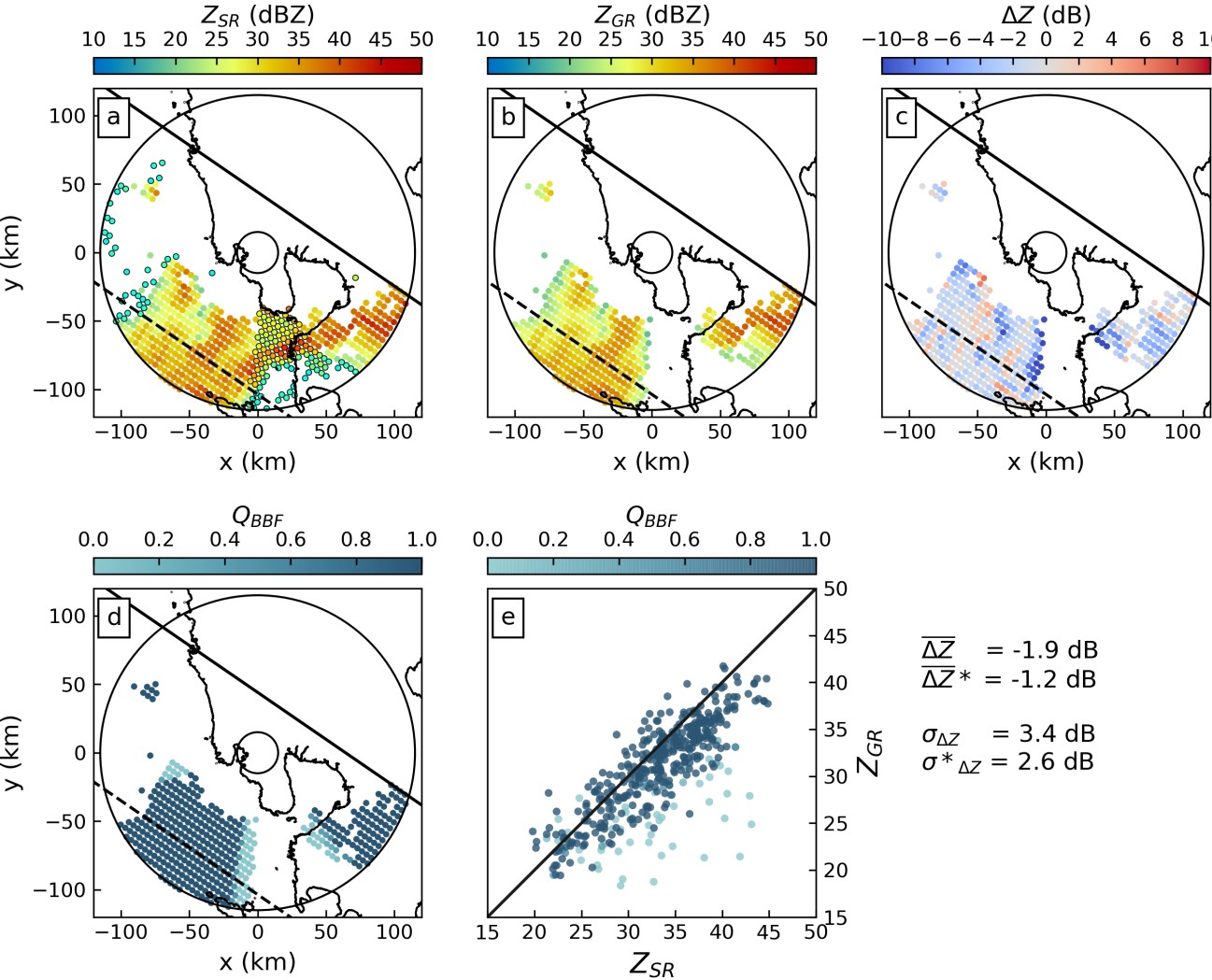

**Figure 5.** GR-centered maps of volume-matched samples from 8 November 2013 at 0.5° elevation angle of (a) SR reflectivity, (b) GR Reflectivity, (c) difference between GR and SR reflectivities, and (d) $Q_{BBF}$. (e) Scatter plot of $Z_{GR}$ vs $Z_{SR}$ where each point is colored based on the data quality ($Q_{BBF}$). The solid line in (a)–(d) is the edge of the SR swath, the other edge lies outside the figure. The dashed line denotes the central axis of the swath. The solid concentric circles demarcate the 15 km and 115 km ranges from the radar. In (a) observations that are present in the SR data but not detected by the GR are encircled in black. The mean bright band is at a height of 4685 meters.

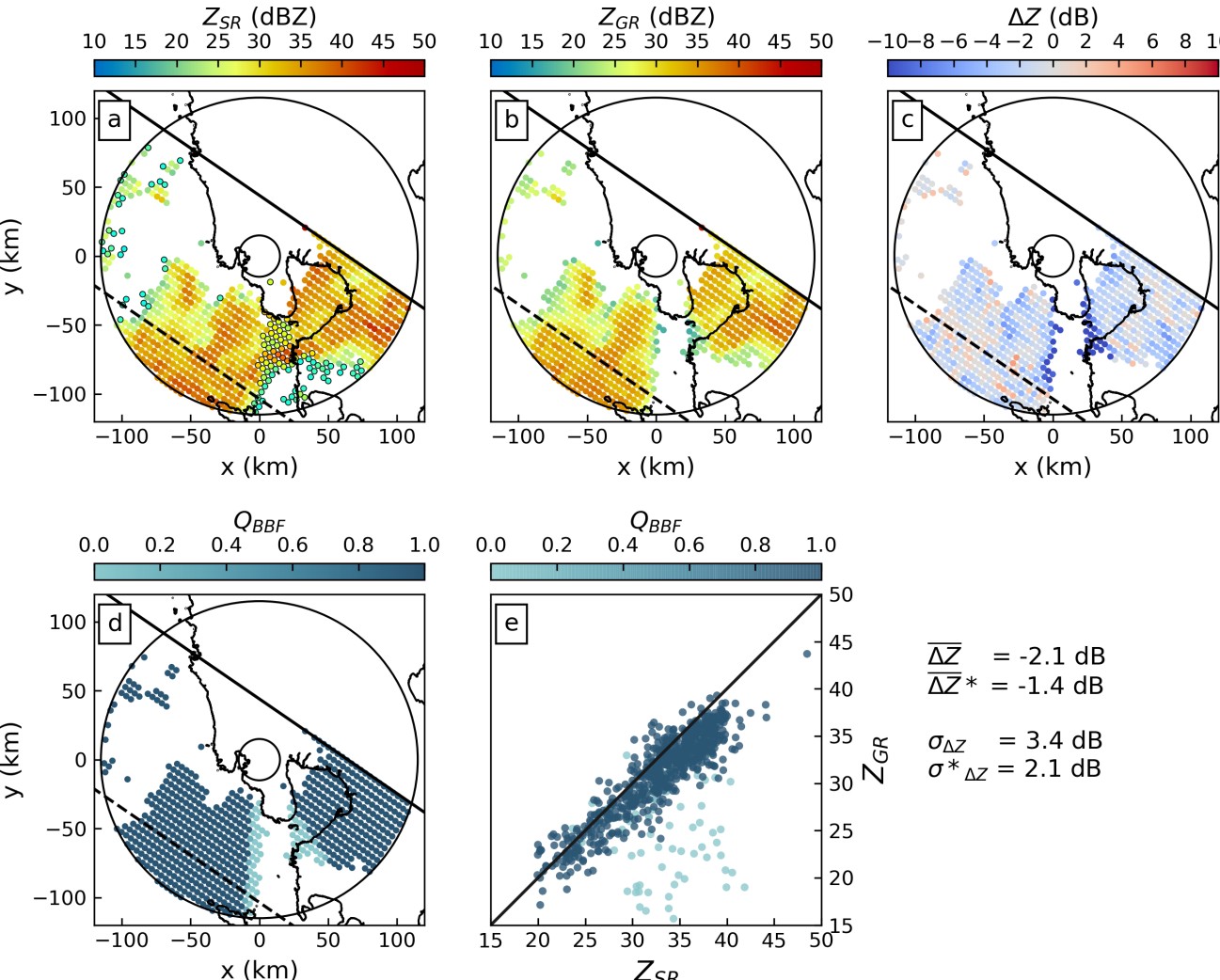

$$\overline{\Delta Z} \quad = \text{-2.1 dB}$$
$$\overline{\Delta Z}* = \text{-1.4 dB}$$

$$\sigma_{\Delta Z} \quad = 3.4 \text{ dB}$$
$$\sigma*_{\Delta Z} = 2.1 \text{ dB}$$

**Figure 6.** Same as in Figure 5 but for 1.5° elevation angle

becomes obvious from Figure 2. Considering these bins in the matched samples will cause a systematic error in the estimate of
calibration bias, unless we use the partial beam blockage fraction as a quality filter by computing a quality-weighted average
of reflectivity. As a consequence, the effect of quality-weighted averaging (with partial beam blockage fraction as a quality
variable) can be most pronounced at "intermediate" elevation angles, depending on the specific topography and its location
with respect to the ground.

The effect becomes obvious for the next elevation angle. Figure 6 is equivalent to Figure 5, but for an elevation angle of
1.5°: As the sector of total beam blockage shrinks at that elevation, the impact of partial beam blockage on the estimation of
GR calibration bias increases. For an antenna elevation of 1.5°, some bins in areas of partial beam blockage have very large

negative biases (over 20 dB). Ignoring beam blockage for this elevation angle yields a bias estimate of -2.1 dB (simple mean), while the quality-weighted average yields a bias of -1.4 dB. At the same time, considering quality substantially reduces the standard deviation from 3.4 dB to 2.1 dB.

### 4.1.2   Case 2: 01 October 2015

The second case confirms the findings in the previous section for a GPM overpass on October 1, 2015. That overpass captured an event in the northern and eastern part of the radar coverage where partial beam blockage is dominant, as well as a small part of the southern sector with partial and total beam blockage. Figure 7 shows the results of that overpass in analogy to the previous figures, for an antenna elevation of 0.0 degree. The figure shows a dramatic impact of partial beam blockage, with a dominant contribution from the northern part, but also clear effects from the eastern and southern sectors. The scatter plot of $Z_{GR}$ over $Z_{SR}$ (e) demonstrates how the consideration of partial beam blockage increases the consistency between GR and SR observations and allows for a more reliable estimation of the GR calibration bias: Ignoring partial beam blockage (simple mean) yields a bias of -2.7 dB, while the quality-weighted average bias is -1.1 dB. Taking into account quality decreases the standard deviation from 3.8 dB to 2.7 dB.

### 4.2   Overall June-November comparison during the 5-year observation period

Finally, we applied both the simple and the quality-weighted mean bias estimation to each of the TRMM and GPM overpasses from 2012 to 2016 that met the criteria specified in Section 3.2, Table 2. As pointed out in Section 3.2, the matching procedure itself is carried out per GR sweep, i.e. separately for each antenna elevation angle.

As a result, we obtain a time series of bias estimates for GR calibration, as shown in Figure 8. In this figure, the calibration bias for each overpass is computed from the full GR volume, i.e. including matched samples from all available antenna elevations. In the upper panel (a), each marker represents the quality-weighted mean bias for a specific SR overpass (circles for GPM, triangles for TRMM). The center panel (b) highlights the differences between the quality-weighted and the simple mean approach, by quantifying the effect of taking into account GR data quality (in this case, partial beam blockage). The bottom panel (c) shows the differences between the quality-weighted standard deviation and the simple standard deviation of differences, illustrating how taking into account GR quality affects the precision of the bias estimates.

The time series provide several important insights:

(1) **Effect of quality weighting on bias estimation**: Figure 8b and c together illustrate the benefit of taking into account GR data quality (i.e. beam blockage) when we estimate GR calibration bias. It does not come as a surprise that the difference between $\Delta Z^*$ and $\Delta Z$ is mostly positive because the areas suffering from partial beam blockage register weaker signals (i.e. lower reflectivity) than expected, producing a lower mean bias. Giving the associated volume-matched samples low weights in the calculation of the mean bias brings the quality-weighted bias up. In the same vein, the beam-blocked bins introduce scatter, and assigning them low weights decreases the standard deviation. Figure 8c shows, as a consequence, that the quality weighted bias estimates are consistently more precise: in the vast majority of overpasses, the quality weighted standard deviation is substantially smaller than the simple standard deviation. That result is also consistent with the case study result shown above.

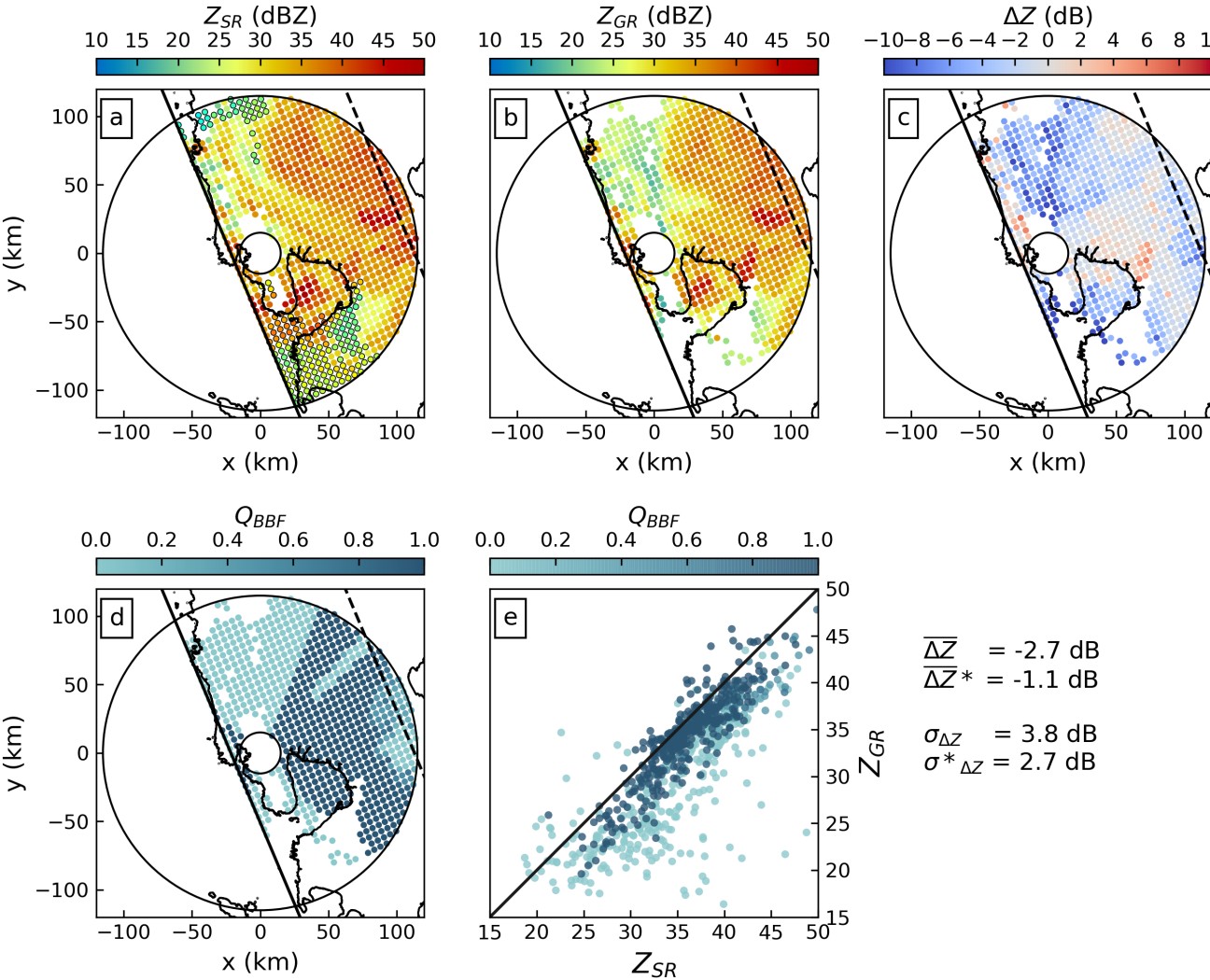

**Figure 7.** As in Figure 5 but for the overpass 01 October 2015. The mean bright band level is found at 4719 meters for this case

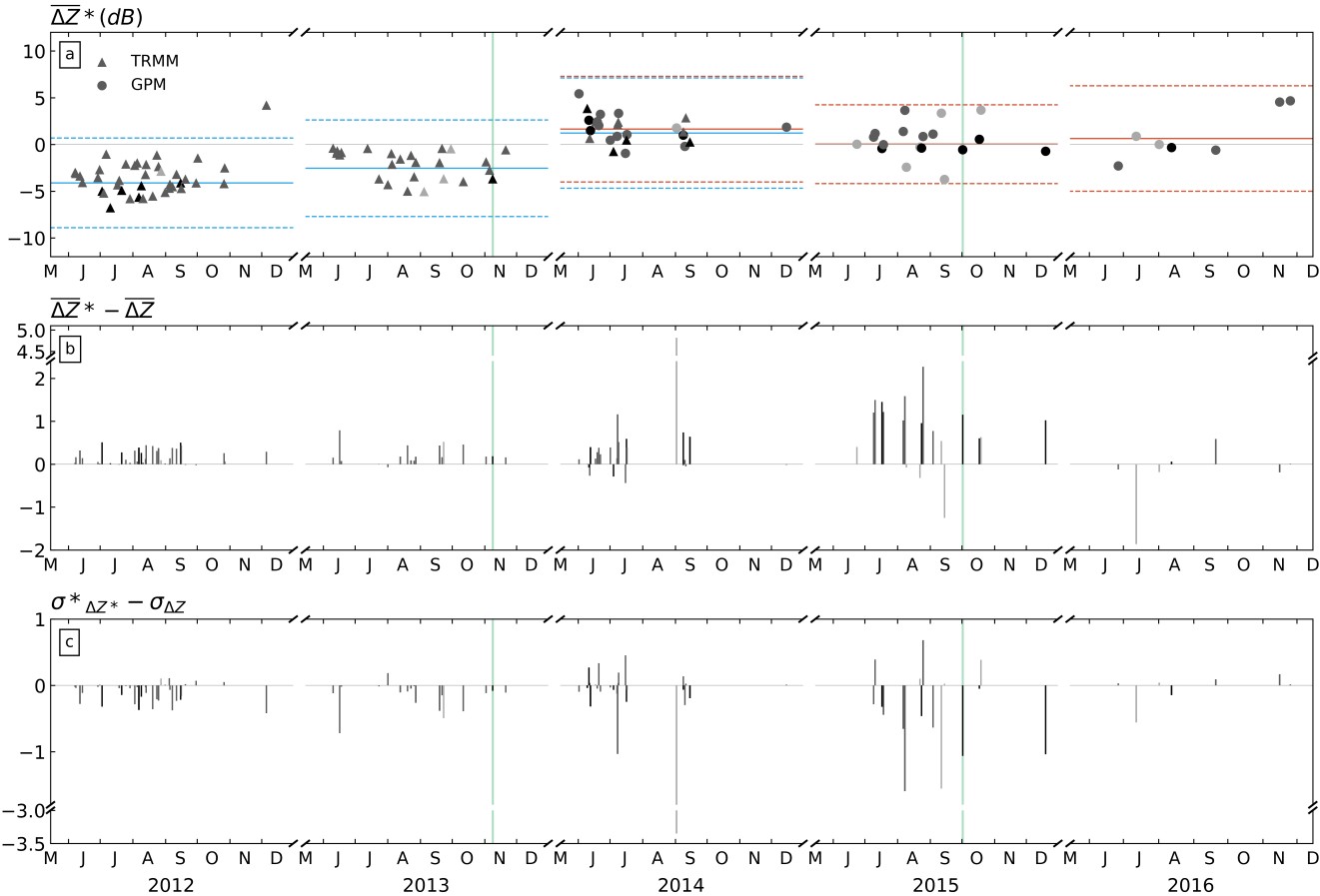

**Figure 8.** (a) Time series of the weighted mean bias ($\Delta Z^*$) from 2012 to 2016. Analysis covers only the wet season from June to December. Triangle markers represent TRMM overpasses while circle markers are GPM overpasses. Symbols are colored according to the number of volume-matched samples on a logarithmic scale: light gray = 10–99, medium gray = 100–999, and black = 1000+. Blue and orange solid (dashed) horizontal lines represent the weighted average (standard deviation) of *all* individual matched samples within the year for TRMM and GPM, respectively. (b) The difference between the weighted mean biases ($\Delta Z^*$) and the simple mean biases ($\Delta Z$). (c) The standard deviation of the weighted mean bias minus the standard deviation of the simple mean bias values. The green vertical lines indicate the date of the two case studies.

It should be noted, though, that for some overpasses, the quality weighting procedure (which is in effect a filtering) can cause an increase in the bias estimate and/or the standard deviation of that estimate. That effect occurs for overpasses with particularly low numbers of matched samples, and, presumably, with rainfall in regions in which our estimated beam blockage fraction is subject to higher errors (caused by e.g. the inadequateness of the assumed Gaussian antenna pattern, variability of atmospheric refractivity, or errors related to the DEM, its resolution and its interpolation to ground radar bins). In total, however, the effect of decreasing standard deviation vastly dominates.

(2) **GPM and TRMM radars are consistent**: In 2014, both TRMM and GPM overpasses are available. That period of overlap shows that the GR calibration bias estimates that are based on both TRMM and GPM observations can be considered as homogeneous. Using TRMM data, the average calibration bias for all 2014 overpasses amounts to $1.6 \pm 1.3$ dB, while using the GPM overpasses yields a bias of $1.8 \pm 1.5$ dB. The difference between TRMM version 7 and GPM version 5 reflectivities mentioned in Section 2.1 falls within the uncertainties in the annual estimated mean bias, which makes us confident that the substantial year-to-year changes of our bias estimates are based on changes in GR calibration.

(3) **Change of bias over time**: Despite the variability of bias estimates between the individual overpass events, the time series still provides us with a clear signal: The bias estimates appear to fluctuate around an average value that appears to be quite persistent over the duration of the corresponding wet seasons of the different years, so over intervals of several months. Considering the average calibration bias over the different wet seasons (horizontal lines in Figure 8a), we can clearly observe changes in calibration bias over time. The bias was most pronounced in 2012 and 2013, with average bias estimates around -4.1 dB for 2012 and -2.5 dB for 2013. For 2014, the absolute calibration bias was much smaller, at a level of 1.4 dB while for 2015 and 2016, the situation improved further, with an average bias of 0.0 dB in 2015 and 0.6 dB in 2016. It is important to note that these values were computed as the average bias and its standard deviation across all matched volumes and not as the average of bias estimates across overpasses. Accordingly, the standard deviation (as indicated by the dashed lines) is quite high since it includes all the scatter from the individual overpasses. We have to assume that a fundamental issue with regard to calibration maintenance was addressed between 2013 and 2014 in the context of hardware changes (i.e. replacement of magnetron). Unfortunately, we were not able to retrieve detailed information on maintenance operations that might explain the changes in bias of the radar throughout the years.

**Short term variability of bias estimates between overpasses**: There is a strong variability of the estimated calibration bias between overpasses (Figure 8a) and spatially within each overpass (Figures 5 to 7). That variability is clearly not a desirable property, as we would not expect changes in calibration bias to occur at the observed frequency, amplitude, and apparent randomness. As a consequence, we have to assume that the variability is a cumulative result of various and dynamic sources of uncertainty along the entire process of observation, product generation, matching, and filtering. That assumption is well in line with many other studies (such as Anagnostou et al. (2001); Durden et al. (1998); Joss et al. (2006); Kim et al. (2014); Meneghini et al. (2000); Rose and Chandrasekar (2005); Schwaller and Morris (2011); Seto and Iguchi (2015); Wang and Wolff (2009); Warren et al. (2018), to name only a few) which discuss e.g. fundamental issues with the backscattering model for different wavelengths and sampling volumes; the uncertainty of beam propagation subject to fluctuations in atmospheric refractivity; residual errors in the geometric intersection of the volume samples; uncertainties in SR reflectivity subject to the effects of

attenuation correction at Ku band, non-uniform beam filling and undesirable synergies between the two; rapid dynamics in backscattering target during the time interval between SR overpass and GR sweep; effects non-meteorological echoes for both SR and GR; and, presumably, also short-term hardware instabilities. Considerung these uncertainties, together with the fact that the quality weighting in our case study explicitly accounts for beam bloackage only, the short term variability becomes plausible. Yet, it is beyond the scope of this study to disentangle the sources of this variability.

## 5    Conclusions

In 2011, Schwaller and Morris presented a new technique to match spaceborne radar (SR) and ground-based radar (GR) reflectivity observations, with the aim to determine the GR calibration bias. Our study extends that technique by an approach that takes into account the quality of the ground radar observations. Each GR bin was assigned a quality index between 0 and 1, which was used to assign a quality value to each matched volume of SR and GR observations. For any sample of matched volumes (e.g. all matched volumes of one overpass, or a combination of multiple overpasses), the calibration bias can then be computed as a quality-weighted average of the differences between GR and SR reflectivity in all samples. We exemplified that approach by applying a GR data quality index based on the beam blockage fraction, and we demonstrated the added value for both TRMM and GPM overpasses over the 115 km range of the Subic S-band radar in the Philippines for a five year period.

Although the variability of the calibration bias estimates between overpasses is high, we showed that taking into account partial beam blockage leads to more consistent and more precise estimates of GR calibration bias. Analyzing five years of archived data from the Subic S-band radar (2012-2016), we also demonstrated that the calibration standard of the Subic radar substantially improved over the years, from bias levels around -4.1 dB in 2012 to bias levels of around 1.4 dB in 2014 and settling down to a bias of 0.6 dB in 2016. Of course, more recent comparisons with GPM are needed to verify that this level of accuracy has been maintained. Case studies for specific overpass events also showed that the necessity to account for partial beam blockage might even increase for higher antenna elevations. That applies when sectors with total beam blockage (in which no valid matched volumes are retrieved at all) turn into sectors with partial beam blockage at higher elevation angles.

Considering the scatter between SR and GR reflectivity in the matched volumes of one overpass (see case studies), as well as the variability of bias estimates between satellite overpasses (see time series), it is obvious that we do not yet account for various sources of uncertainties. Also the simulation of beam blockage itself might still be prone to errors. Nevertheless, the idea of the quality-weighted estimation of calibration bias presents a consistent framework that allows for the integration of any quality variables that are considered important in a specific environment or setting. For example, if we consider C-band instead of S-band radars, path-integrated attenuation needs to be taken into account for the ground radar, and wet radome attenuation probably as well (Austin, 1987; Merceret, 2000; Villarini and Krajewski, 2010). The framework could also be extended by explicitly assigning a quality index to SR observations, too. In the context of this study, that was implicitly implemented by filtering the SR data e.g. based on bright band membership. An alternative approach to filtering could be weighting the samples based on their proximity to the bright band, the level of path-integrated attenuation (as e.g. indicated by the GPM 2AKu

variables *pathAtten* and the associated reliability flag (*reliabFlag*)) or the prominence of non-uniform beam filling (which could e.g. be estimated based on the variability of GR reflectivity within the SR footprint, see e.g. (Han et al., 2018)).

In addition, with the significant effort devoted to weather radar data quality characterization in Europe (Michelson et al., 2005), and the number of approaches in determining an overall quality index based on different quality factors (Einfalt et al., 2010), it is straightforward to extend the approach beyond beam blockage fraction.

Despite the fact that there is still ample room for improvement, our tool that combines SR–GR volume matching and quality-weighted bias estimation is readily available for application or further scrutiny. In fact, our analysis is the first of its kind that is entirely based on open source software, and thus fully transparent, reproducible and adjustable (see also Heistermann et al. (2014)). Therefore this study, for the first time, demonstrates the utilization of wradlib functions that have just recently been implemented to support the volume matching procedure and the simulation of partial beam blockage. We also make the complete workflow available together with the underlying ground and space-borne radar data. Both code and results can be accessed at the following repository `https://github.com/wradlib/radargpm-beamblockage` upon the publication of this manuscript.

Through these open-source resources, our methodology provides both research institutions and weather services with a valuable tool that can be applied to monitor radar calibration, and—perhaps more importantly—to quantify the calibration bias for long time series of archived radar observations, basically beginning with the availability of TRMM radar observations in December 1997.

*Code and data availability.* Code and sample data can be accessed at `https://github.com/wradlib/radargpm-beamblockage`

*Competing interests.* The authors declare that they have no conflict of interest.

*Acknowledgements.* The radar data for this analysis were provided by the Philippine Atmospheric, Geophysical and Astronomical Services Administration (PAGASA, http://pagasa.dost.gov.ph). The study was also funded by the German government through the German Academic Exchange Service (https://www.daad.de/en/).

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
