# Peer review of "Enhancing the consistency of spaceborne and ground-based radar comparisons by using beam blockage fraction as a quality filter"

_Atmospheric Measurement Techniques, 2018_

## Short Comment (SC1) · 12 May 2018

D. Michelson

dbm@baltrad.eu

General comments This is a very useful combination of volume matching and quality characterization with the additional benefit that the solution is implemented in open source software. There are, however, some issues that require attention before the paper is in publishable form.

Specific comments I can identify three fundamental aspects of this paper that require clarification and elaboration: 1. How to characterize data quality? Significant effort has been devoted to systematic representation of weather radar data quality in Europe, through COST Actions (717, 731) and EUMETNET OPERA, giving framework

approaches for dealing with data quality. The authors have followed Zhang et al. (2011) for representing data quality resulting from beam blockage. It would be useful to have some context in the paper acknowledging previous work and including a rationale for choosing the Zhang et al. approach. 2. What advantages does this work offer when it comes to addressing topographical beam blockage with GR compared to previous work? The paper references Bech et al. (2003) which is a benchmark paper. There are other implementations of the same approach that use other DEM data, e.g. GTOPO30. The authors' use of high-resolution SRTM data is interesting, but are the results better than using ∼1 km GTOPO30 data? 3. GR calibration. There is one sentence in 3.2(2) indirectly indicating that the Subic radar may have been calibrated during the time period covered by the study. This needs to be clarified. Was the radar calibrated during this time? More than once? Are the results available? Any other maintenance that could have impacted calibration levels? This is very important to understand results like those presented in Figure 8. Also, the methods presented in this paper have been applied to data from one GR, yet they would be much more valuable if also applied to data from a second GR. Doing so would reveal which radar is "hot" and which is "cold" and whether there are any other systematic differences that are unique to each GR.

Technical "corrections" Including information throughout the paper on what software calls are available and have been used is irrelevant and should be avoided. Instead, I recommend a small section following the recommendations given by Irving: https://doi.org/10.1175/BAMS-D-15-00010.1

References to Morris and Schwaller and Schwaller and Morris are inconsistent. In the list of references, both are given from 2011, but the paper references one from 2009. The Morris and Schwaller reference appears to be incomplete in the list of references.

2.1.2 Version 6 of the GPM 2AKu products is stated, but is not the current version 05B? Reference(s) to product documentation are needed. 2.3 Where does the information on bright-band height and width come from? Also precipitation type and rain indicators? Please add.

2.3 GR data are acquired every 9 minutes, but matched within a 5-min window. How is this done?

Figures 5-7. Sub-plot (e) is a great way to visualize this kind of result, but clearer colours are needed. I'm suspecting that light-gray points are covered by dark-grey points. A colour table might be a better approach, perhaps combined with slightly smaller point sizes.

Figure 5 caption: Replace ZPR with ZSR

Figure 8. Might want to clarify in the caption that data from Jan-Apr are not used because this is the dry season.

Just a thought: what impact can radome wetting have on the results? Radome wetting is still an issue even if you exclude data near the radar. But is it an issue at all at S band?

Review rating Scientific significance: 2 Scientific quality: 3 Presentation quality: 2

---

## Referee Comment (RC1) · M Gabella (Referee) · 22 May 2018

**A review of "Enhancing the consistency of spaceborne and ground-based radar comparisons by using quality filters" by Irene Crisologo et al.,** submitted to AMT

The paper presents a large data set (5 years of data, 283 overpasses downloaded) and interesting comparison between a single ground-based radar (GR) and two satellite precipitation radars (SPR):

- the first ever spaceborne precipitation radar (PR) on-board the TRMM satellite (183 overpasses from June 1, 2012 to September 30, 2014).
- the follow-on of TRMM PR on board the GPM core satellite (103 overpasses, from June 1, 2014 to November 2016). GPM carries a dual-frequency Ku- Ka- band radar (DPR) rather than the single-frequency Ku-band PR on board TRMM.

Many efforts have been made to provide spaceborne radars with long-term electronic stability. The calibration factor has assumed to have an accuracy better than 1 dB. As a consequence, the authors have used (quality-weighted) average of geo-located TRMM PR and GPM DPR reflectivity values as reference to check the quality of the reflectivity bias of the S-band GR (Note that the number of overpasses with "enough" reflectivity values reduces to "only" 74 for TRMM and 40 for GPM)

The authors have done a thorough job of comparing (quality-)averaged reflectivity of the GR versus the satellite one. Hence, I find the paper valuable and interesting: it is another example of weather radar monitoring and reflectivity bias assessment using the SPR as reference. It shows once more the importance of monitoring (calibrate and adjust) GRs reflectivity before a quantitative use for precipitation estimation. I think it is particularly valuable and illustrative for National Weather Services worldwide. However, I feel that there are some issues that need to be specifically addressed in the text/Tables before the paper becomes in publishable form. Schematically, I have identified four points, which will be described below

1) Omission (or insufficient description) in the text of limiting factors in the quantitative interpretation of radar reflectivity.
2) GR monitoring based on SPR: possible causes of the discontinuity affecting the GR?
3) (Radar reflectivity) Mean Field Bias definition and assessment.
4) Literature

I also propose a different structure for the paper: Sec. 2 Data; Sec. 3 Methods; Sec. 4 Results and Discussion (see page 4 for details)

I would also suggest the make the final part of the title more specific, for instance " … by using a quality filter based on beam blockage fraction"

Yours Sincerely,
Marco Gabella

Locarno-Monti, 22.5.2018

1)
I hope the authors can agree with the following three considerations:
a) Quantitative interpretation of radar measurements are based on A MODEL of the backscattering targets.
b) Such A MODEL is an approximation of a very complex reality (Nature).
c) There is never sufficient information in radar measurements to resolve such complexity.

Having said that, I think I can now recommend more emphasis in the text related to the very different wavelengths and sampling volumes (for instance, you may want to have a look at the figures in the paper by Joss et al., 2006) characterizing GR (3 GHz) vs SPR (14 GHz, attenuating frequency!). Yes, one can try to correct for attenuation (e.g., Iguchi et al., 2000), he can even try to

convert Z from 10 to 2 cm, but the uncertainties affecting the retrieved quantities are large! (See a) b) c) above … )

By the way, when introducing Eq. (2), you mention Cao et al. (2013) and coefficients in Table 1 for dry snow and hail … I have just quickly opened the pdf and saw that Table 1 lists (retrieval/ simulated) BIAS and RMSE?!?

I am confident that after (re-)considering the above mentioned issues, after thinking of the (necessarily) simplifying approach for beam occultation correction[1] (Gaussian shape for the main lobe of the antenna radiation pattern, instead of the simple and practical linear approach proposed by Bech et al., which is an unrealistic "top-hat" radiation pattern), …, the authors will feel more comfortable with what they call "short term variability" at page 15, line 7; furthermore, they will not list "short term variability" at the first place, rather … at the last one!

2)

What happened to the GR in 2014? (lines 20-22, page 15 and Figure 8): +1.4 dB overestimation, after two year of clear and "heavy" under-estimation! (–4.1 dB and –2.5 dB, respectively). What a jump! Was it hardware related? Software? Both? From a weather service viewpoint, it is interesting that this paper bring in the important concept of GR calibration and monitoring, see e.g. the recent successful workshop

https://www.dwd.de/EN/specialusers/research_education/seminar/2017/wxrcalmon2017/wxrcalmon_en_node.html

However, if the authors provided possible explanations of what happened, the paper would become even more interesting and valuable. If you are interested in knowing more regarding monitoring and calibration of modern radar, you may find recent paper regarding: the Transmitter chain (e.g., Reimann et al., 2016); the Receiver chain (for instance, using the Sun: Gabella et al., 2016; Hubbert, 2017,) both Transmitter and Receiver chains, using a 24 GHz vertically pointing radar and disdrometers (Frech et al., 2017).

Gabella, M.; Boscacci, M.; Sartori, M.; Germann, U. Calibration accuracy of the dual-polarization receivers of the C-band Swiss weather radar network. Atmosphere, 2016, 7, 76.

Reimann, J.; Hagen, M. Antenna pattern measurements of weather radars using the Sun and a point source. J. Atmos. Ocean. Technol. 2016, 33, 891–898.

Frech, M.; Hagen, M.; Mammen, T. Monitoring the absolute calibration of a polarimetric weather radar. J. Atmos. Ocean. Technol. 2017, 34, 599–615.

However, maybe, using a more robust definition for the SPR-GR reflectivity Bias, it will come out that the jump is smaller than 3.9 dB; for what concerns the assessment of the Mean Field Bias and its statistical evaluation, please have a look at following point #3)

3)

From my viewpoint, the study is a bit limited in the definition of Bias assessment and the corresponding statistical metrics for the evaluation. For instance, it is going to be straightforward for the authors to derive other statistical parameters and present them in a summarizing Table that can complement the nice and informative figure 8. First of all, in addition to the annual mean of $\{\Delta Z_{dB}*\}$ (lines 20-22, page 15) also the standard deviation of $\{\Delta Z_{dB}*\}$. Then, I would suggest a more robust definition for the Bias: instead of using dBZ, you use Z values in linear units: $Z=10^{(dBZ/10)}$. Then you derive a weighted average for the numerator (denominator) using linear Z of the GR (SPR). Finally, you compute 10 Log of such ratio (dB). This annual Log_of_the_MFB is more resilient than the Mean_of_the_Log presented in the paper.

To avoid weighted-average or in a probability matching scheme, you may want to consider only bins with $Q_{BBF}$ larger than say, 0.9 (or larger). After having done this selection, consider the difference ($BIAS_{xx}$) between different quantiles (probability matching): xx=50 (median), 75, 84, 90, 95, 99 percentiles. Maybe, $BIAS_{xx}$ is not constant, rather it depends on the percentile? Finally, for
* * *
[1] Bech et al. wrote " … less idealized, but more realistic, is the case of a Gaussian beam."

these $Q_{BBF}$-selected bins, you may explore the value of the average bias $E\{\Delta Z_{dB}\}$ as a function of the intensity of the echo of the GR (using for instance intervals of 3 or 5 dBZ; obviously, you will have less and less samples for larger values of $dBZ_{GR}$). Does this Mean_of_the_Log Bias remain more or less constant? Or do you see a trend? (Maybe, SPR has residual attenuation for large reflectivity values?). Interesting, is not it?

4)

Finally, the last issue is related to literature: while several TRMM PR vs GR papers are listed, there is a lack of DPR-related studies and DPR technical literature. regarding the latter, I have suggested at the end (GPM related references), three papers published in 2014 and 2015. Regarding the former, I have listed our recent DPR-related studies in the complex terrain of Switzerland; I am confident the authors will be able to find additional GPM papers also in other parts of the world. Furthermore, is Cao et al. double citation (at page 6) correct? Does Morris and Schwaller (2009) exist? (line 7, page 1 citation)

Regarding GPM, please, do not forget to mention that your analysis neglect Ka-band observations (please briefly discuss the reason of such a choice).

ADDITIONAL SUGGESTED REFERENCES

Iguchi, T., T.Kozu, R. Meneghini, J.Awaka, and K. Okamoto, "Rain-profiling algorithm for the TRMM Precipitation Radar," J. Appl. Meteorol., vol. 39, pp. 2038–2052, 2000.

Keenan, T., E. Ebert, V. Chandrasekar, V. Bringi, and M. Whimpey (2003), Comparison of TRMM satellite-based rainfall with surface radar and gauge information, paper presented at 31st Conference on Radar Meteorology, Am. Meteorol. Soc., Seattle, Wash., 6–12 Aug.

Joss, J., et al, 2006: Variation of weather radar sensitivity at ground level and from space: Case studies and possible causes, Meteorol. Z., 15, 485–496, doi:10.1127/0941-2948/2006/0150.

GPM related references

Kubota, T., and Coauthors, 2014: Evaluation of precipitation estimates by at-launch codes of GPM/DPR algorithms using synthetic data fromTRMM/PRobservations. IEEEJ. Sel. Top.Appl.Earth Obs. Remote Sens., 7, 3931–3944, doi:10.1109/JSTARS.2014.2320960.

Furukawa, K., T. Nio, T. Konishi, R. Oki, T. Masaki, T. Kubota, T. Iguchi, and H. Hanado, 2015: Current status of the dual-frequency precipitation radar on the global precipitation measurement core spacecraft. Proc. SPIE, 9639, 96 390

Toyoshima, K., H. Masunaga, and F. A. Furuzawa, 2015: Early evaluation of Ku- and Ka-band sensitivities for the Global Precipitation Measurement (GPM) Dual-Frequency Precipitation Radar (DPR). SOLA, 11, 14–17, doi:10.2151/sola.2015-004.

Speirs et al., 2017: A Comparison between the GPM Dual-Frequency Precipitation Radar and Ground-Based Radar Precipitation Rate Estimates in the Swiss Alps and Plateau, *J. Hydrometeorol.*, DOI: 10.1175/JHM-D-16-0085.1

Gabella et al., 2017: Measurement of Precipitation in the Alps Using Dual-Polarization C-Band Ground-Based Radars, the GPM Spaceborne Ku-Band Radar, and Rain Gauges. *Remote Sensing*, 9, 1147; 19 pp.; doi:10.3390/rs9111147

**Specific Suggestions**

Introduction

-Line 4: … to monitor the **bias of the gauge adjustment factor to be applied to precipitation estimates** of the GR.

Lines 6: … to quantify the GR **reflectivity** bias with respect to the reference (namely, SR reflectivity value after conversion from Ku-band to S-band).

In fact, I would propose the following terminology:

Setting the Bias as close as possible to 0 dB between radar QPE and in situ measurements: → gauge-adjustment

Assessing the Bias between reflectivity of two radars: → relative calibration

Forcing to 0 dB the Bias in measured Power (dBm) between an external or internal reference Noise Source and the radar at hand: → absolute calibration

Section 2.1.2

The fact that only 283 overpasses were within the selected, reasonable 120 km range should be mentioned here. There is no reason to wait until former(see **) sec. 3.1.1. Similarly, you can at least anticipate that the number will considerably decrease upon conditional requirements such as min. # of "wet" pixels, time difference, min. # of bins above both GR and SPR sensitivity.

Question: have you only used only months from June to November? Not clear from the text. Please rephrase. In fig. 8, I see two overpasses in December (2012 and 2014). By the way, in Dec. 2012 $E\{\Delta Z_{dB}*\}$ is almost 5! dB, while the annual average is -4.1 dB?!? (see my previous points 2) and 3) )

**By the way, I would propose the following structure for Sections and Subsections

2. Data
2.1 Spaceborne Precipitation Radar (SPR)
2.2 GR

3. Method
3.1 Partial beam shielding and quality index based on beam blockage fraction
3.2 SPR-GR volume matching
3.3 Assessment of the average reflectivity Bias

Section 3.1: I would move it (including former fig. 4) inside the new Section 3.1 (former Sec. 2.2)

4. Results and Discussion
4.1 Single event comparison
4.1.1 Case1
4.1.2 Case2
4.2 Overall June-November comparison during the 5-year observation period

Page 5, Line 4-5: Please delete the sentence, the reader is able to read the simple algebra in eq. (1).

Page 10, Lines 19-25: misleading. I cannot possibly agree. On the contrary, my interpretation is that partial beam blockage plays approximately the same role (0.7 dB difference between the silly estimate that include blockage and the conservative one that exclude all cases where BBF > 0.5). Please rephrase.

Page 13: Would you please add a complementary figure at ELEV= 1.5 for the 1.10.2015 overpass? Just like you did for the 8.11.2013 overpass.

Page 14 and Figure 8. Some journals ask for a graphical abstract as a self-explanatory image to appear alongside with the abstract. I think Fig. 8 would be perfect for such scope. It is nice and rich of information. Suggestion: could you please use color. For instance, the 1.10.2015 and 8.11.2013 overpasses could be in color. By the way, the 8.11 circle in picture a) seems to be very close to 0 dB, while in Fig. 7 it is written that $E\{\Delta Z_{dB}*\}$ is -1.1 dB. Am I missing something? Is it related to what you wrote in lines 3-6? These sentences are not clear to me, could you rephrase, please?

Furthermore, regarding picture b), do not forget to emphasize that if the $Q_{BFF}$ works properly then: $E\{\Delta Z_{dB}*\} - E\{\Delta Z_{dB}\}$ should be negative in 2012 and 2013 (almost all the point in a) are below the 0 dB dotted line), positive in 2014 (almost all the point in a) above the 0 dB dotted line …).

Page 15. I would change the order of your points and list your point (1) at the end, as # (4) [see my comment 1) at page 1)]. I would start from (3), which is the scope of this paper: indeed an intelligent weighted-average based on $Q_{BFF}$ shows a better standard deviation of $\Delta Z_{dB}*$. By the way, I recommend adding a table and/or a figure (histogram) that summarizes the statistical properties of $\sigma*\{\Delta Z_{dB}*\}$ and $\sigma\{\Delta Z_{dB}\}$.
Then, I would introduce the important result regarding the consistency of GPM and TRMM radars, followed by the changes of the bias in time

Page 16.
Line 5, delete coherent.
Line 14-16. Sorry, you cannot summarize the (mis-) calibration of the GR by simply going from 2012 (–4.1 dB) to 2016 (+0.6 dB) and omit, for instance, the +1.4 jump in 2014. [see my comment 2) at page 2)].
Line 17-19. Pleonastic. I would delete it.
Line 26. Why do you discuss C-band radar technology ?

Minor points

My proposal for radar acronyms: 2-character for ground, namely GR; 3-character for satellite radar. Would you please use TPR for TRMM, DPR for GPM and SPR in those cases where you refer to both, independently of the platform?

---

## Referee Comment (RC2) · Anonymous Referee #2 · 23 May 2018

**Enhancing the consistency of spaceborne and ground-based radar comparisons by using quality filters**

Irene Crisologo et al.

**Review by:**

Anonymous Referee

**General Comments:**

The authors provide an analysis of their technique to assess the calibration bias of a ground-based radar against TRMM/GPM in the Philippines. Their contribution to the scientific community is not only a peer-reviewed technique (Schwaller and Morris 2011) applied to a new dataset, but also a quality control procedure to assist with uncertainty caused by beam blockage from terrain and other terrestrial objects. This is useful to the scientific community because of the recent popularity of comparing GR to SR (e.g., 3 papers published this year: Warren et al. 2018, Han et al. 2018, Zhang et al. 2018) which could improve calibration results by including the quality-weighted average suggested by the authors here. Furthermore, the added value of the software being open-source allows for future investigations to implement this methodology easily and quickly.

It is suggested that the paper be accepted with minor revisions after addressing the following concerns:

**Comments on other sources of uncertainty in calibration assessment:**

1) *Attenuation at Ku-band:*

The authors should address the uncertainties with attenuation correction at Ku-band. The attenuation correction tech. used for just Ku-band is the HB-SRT method (Seto and Iguchi 2015). It is known that using the HB method alone does not work well in higher rain rates (> 20 mm hr$^{-1}$, Seto and Iguci 2011, but as low as 12 mm hr$^{-1}$ Rose and Chandrasekar 2005). Furthermore, the SRT method is more uncertain over land (larger standard deviation of the surface backscatter cross-section, Meneghini et al. 2000). It is anticipated that since the radar is located in the tropics both of the issues above could occur (more likely in convective precipitation). Please discuss these uncertainties and how they could impact your results of the bias correction.

It is mentioned in the conclusions that for C-band attenuation correction is vital, but GPM and TRMM are Ku-band, thus isn't it vital as well?

2) *Ground Clutter for the SR:*

In radar gates near the surface, with respect to the SR, ground clutter is a problem. How are the authors dealing with ground clutter from the SR? Are they using gates below the lowest clutter free bin estimate (included in the GPM file)? If so, is the lowest clutter free gate being

assigned to all the gate below it? If you plot it out, a lot of times that's what is done. Essentially the data looks smeared from the lowest clutter free bin to the surface, which isn't to realistic and it is suggested to just not consider these gates. Please comment on this, potentially in Section 2.3. If you are including these interpolations, you may wish to not (it will introduce error).

3) *NUBF:*

Please also include some discussion of the potential impacts of non-uniform beam filling (NUBF) on your analysis. Edges of large systems, individual cumulus showers could result in NUBF in SR because of the quasi-large footprint. Lowering the reflectivity value in the gate.

**Specific Comments:**

1) Page 2, line 5: Please add the Kummerow et al. (1998) paper for TRMM, and the Hou et al. (2014) for the GPM reference (page 2, line 6). This will help readers who are not entirely familiar with both platforms.

2) Page 6, line 3: "The gates below and above the brightband were considered in the comparison". Please provide a brief reason why this is done. I do not want to assume the authors reasoning.

3) Figure 4 & Section 3.1: It is not clear what you are plotting. The figure titles state the quality index but the figure caption and text states beam blockage fraction. Please clarify.

4) Section 3.1.1: Why are the number of overpasses here different than when they were listed earlier (section 2.1.2)? I am referring to the numbers before applying the criteria in Table 2.

5) Case studies (Section 3.1.2 and 3.1.3): Could you include the mean BB level height? You can add it to the bottom right with the other statistics. Also comment on fraction of stratiform vs convective. These two will help readers assess the amount of attenuation and NUBF that could be involved (e.g. uncertainty in the SR measurements).

6) Figure 5 + 6 + 7 a and b: Suggestion. Consider changing the colorscale to one that is perceptually uniform and color-deficient friendly. For example, try the HomeyerRainbow or the LangRainbow included in Pyart (https://github.com/ARM-DOE/pyart)

7) Page 10, Line 12: "Major parts of that sector did not receive any signal due to total beam blockage". Where is this occurring? The reader can refer back to Figure 4, but it might be helpful to outline the circles with a thin black line in Figure 5d where there is SR data, but no GR data. That way the readers would see where there is 100% beam blockage and thus no signal from the GR, but also gain insight of size of the precipitating system.

8) Page 10, Line 24-25: "That might be considered counterintuitive, as one might expect the blockage to disappear with higher elevations". Please provide some discussion explaining why this is the case.

9) Page 16, Lines 13 – 16. 'We could' and 'we could also' imply that you did not conduct this analysis when it seems you have. I suggest to change these phrases to be definitive. 'We showed that…' 'we also demonstrated that…'

**Technical corrections:**

0) Page 3, line 20: The most current GPM version is version 5, version 6 is not released yet.

1) Page 18, line 18: Reference Cao et al. 2013 is incorrect. It should be:
   a. Empirical conversion of the vertical profile of reflectivity from Ku-band to S-band frequency

2) The reference Warren et al. should be 2018, published Feb 2018 in J. Atmo. + Ocean. Tech.. Page 2, line 8;Page 3,line 25;Page 5,line 11;Page 15,line 14

3) Figure 4: Missing y-ticks and tick labels on bottom left subplot

4) Page 8, line 5-6. No need for new paragraph. You can combine the two.

5) Figure 5: Figure caption has Zpr instead of Zsr

**References:**

Han, J., Chu, Z., Wang, Z., Xu, D., Li, N., Kou, L., Xu, F. and Zhu, Y.: The establishment of optimal ground-based radar datasets by comparison and correlation analyses with space-borne radar data. Meteorological Applications, 25, 161 – 170, 2018.

Hou, A., Kakar, R. K., Neeck, S., Azarbarzin, A. A., Kummerow, C. D., Kojima, M., Oki, R., Nakamura, K. and Iguchi, T.: The global precipitation measurement mission. BAMS, 95, 701 – 722, 2014.

Kummerow, C., Barnes, W., Kozu, T., Shiue, J. and Simpson, J.: The tropical rainfall measuring mission (TRMM) sensor package. Journal of Atmospheric and Oceanic Technology, 15, 809 – 817, 1998.

Meneghini, R., Iguchi, T., Kozu, T., Liao, L., Okamoto, K., Jones, J. A. and Kwiatkowski, J.: Use of the surface reference technique for path attenuation estimates from TRMM precipitation radar. Journal of Applied Meteorology, 39, 2053 – 2070, 2000.

Seto, S. and Iguchi, T.: Applicability of the iterative backward retrieval method for the GPM dual-frequency precipitation radar. IEEE Transactions on Geoscience and Remote Sensing, 49, 1827 – 1838, 2011.

Zhang, S., Zhu, Y., Wang, Z. and Wang, Y.: Consistency analysis and correction of ground-based observations using space-borne radar. Journal of Atmospheric and Solar-Terrestrial Physics, 169, 114 – 121, 2018.

---

## Author Comment (AC1) · 29 Jun 2018

**Final response in the interactive discussion**

Dear Referees,

We would like to thank you for your comments to our manuscript entitled *"Enhancing the consistency of spaceborne and ground-based radar comparisons by using quality filters"* (amt-2018-101). In this document, we would like to provide our responses to the comments of each of the three referees in one single document.

The referee comments turned out to be very helpful. Based on these comments, we suggest several changes to the manuscript which we will outline in detail on the following pages.

For that purpose, we will show the referee comments in black font, and our responses in blue. For the sake of clarity, we have also not reproduced some introductory parts of the referee comments in this comment. Parts that were not reproduced, are marked as *[...]*. Furthermore, we have assigned numbers to all comments to enable cross-referencing between comments. Finally, please find all referenced literature at the end of the response.

We hope that the suggested changes sufficiently address the referees' concerns, so that we can, given the approval of the editor, finalize the revision of our manuscript.

Sincerely,
Irene (on behalf of the author team)

**Short Comment # 1 (by Daniel Michelson)**

[…]

**1)** How to characterize data quality? Significant effort has been devoted to systematic representation of weather radar data quality in Europe, through COST Actions (717, 731) and EUMETNET OPERA, giving framework approaches for dealing with data quality. The authors have followed Zhang et al. (2011) for representing data quality resulting from beam blockage. It would be useful to have some context in the paper acknowledging previous work and including a rationale for choosing the Zhang et al. approach.

We now refer to the framework for quality representation in OPERA, and briefly compare it to the approach of Zhang et al. (2011). It should be noted, though, that our general approach is open to other definitions of overall data quality, as long as such an overall value can be used to compute weights. Furthermore, we used only one single quality variable (beam blockage fraction). Hence, the challenge to aggregate several variables to a single index is not prominent in our study.

**2)** What advantages does this work offer when it comes to addressing topographical beam blockage with GR compared to previous work? The paper references Bech et al. (2003) which is a benchmark paper. There are other implementations of the same approach that use other DEM data, e.g. GTOPO30. The authors' use of high-resolution SRTM data is interesting, but are the results better than using ~1 km GTOPO30 data?

Generally, any increase in DEM resolution should be expected to increase the accuracy of our estimate of the beam blockage fraction (see e.g. Kucera et al. 2004). That effect could be particularly prominent in the near range of the radar as has been shown e.g. for high-resolution airborne laser DEMs by Cremonini et al. (2016). Yet, there is no reference (truth) that could be used to actually verify beam blockage estimates for any underlying DEM. What could be done, however, is to repeat the analysis with GTOPO30 data (i.e. 1 km resolution), in order to investigate the sensitivity of results, or whether our estimate of calibration bias becomes more or less consistent using GTOPO30. Yet, we already consider the paper as quite long, especially considering the requested changes in the course of this review process. We are thus hesitant to include such additional analysis that is not expected to add substantial new insight. Instead, we will add a brief discussion on the potential effects of DEM resolution on the quantification of partial beam blockage.

**3)** GR calibration. There is one sentence in 3.2(2) indirectly indicating that the Subic radar may have been calibrated during the time period covered by the study. This needs to be clarified. Was the radar calibrated during this time? More than once? Are the results available? Any other maintenance that could have impacted calibration levels? This is very important to understand results like those presented in Figure 8. Also, the methods presented in this paper have been applied to data from one GR, yet they would be much more valuable if also applied to data from a second GR. Doing so would reveal which radar is "hot" and which is "cold" and whether there are any other systematic differences that are unique to each GR.

We entirely agree that it would be interesting to apply the methodology to another or even several other radars in order to investigate specific characteristics of individual radars, or in other words, differences and similarities. Yet, we consider this study as a proof-of-concept in which we present the underlying methodology, and show that it adds value in estimating calibration bias for a single radar. An inter-comparison with several radars would be an additional study. Such a study should not only compare the behaviour of different radars independent of each other. It should also investigate the effect of "recalibration" on the consistency between two or more radars in regions of overlap (c.f. Warren et al. 2018). However, we consider such an analysis beyond the scope of the present paper.

Unfortunately, we were unable, despite repeated attempts, to retrieve detailed information on maintenance operations from the radar operator. The only information provided by PAGASA engineers was that in 2012 and 2013, there were problems with the transmitter which caused the output power to be very low (~80 dBm as opposed to the required 89.2 dBm). The modulator was replaced in 2014. In 2015, the magnetron was replaced. In October 2016, the supplier was changed to SELEX. We also have

to assume that after performing major changes in the radar hardware, the radar engineers carried out a calibration.

**Technical "corrections"**

**4)** Including information throughout the paper on what software calls are available and have been used is irrelevant and should be avoided. Instead, I recommend a small section following the recommendations given by Irving: https://doi.org/10.1175/BAMS-D-15-00010.1

In the revised version of the paper, we will discuss the context of Irving (2016). In fact, our paper moves beyond the "minimum standard" suggested by Irving, since we not only provide the code (a doi pointing to the wradlib package version will be generated for the final version of the paper), but we combine the enhancement of an existing software package (wradlib, extensively documented) with a fully documented application context (a jupyter notebook), combined with the data, so that we do not have to provide a log file, as suggested by Irving, but directly allow the user to run the analysis, and modify it in order to adapt to different application contexts. However, we agree that it makes sense to remove the references to explicit function calls from the main text. In order to conform with Irving's suggestions, we also add a brief section with the description of the underlying software, its dependencies, and the notebook to reproduce the results.

**5)** References to Morris and Schwaller and Schwaller and Morris are inconsistent. In the list of references, both are given from 2011, but the paper references one from 2009. The Morris and Schwaller reference appears to be incomplete in the list of references.
The Schwaller and Morris citations will be updated to consistently refer to Schwaller and Morris (2011). The Morris and Schwaller reference will be updated to show complete information.

**6)** 2.1.2 Version 6 of the GPM 2AKu products is stated, but is not the current version 05B? Reference(s) to product documentation are needed.

The version will be corrected to 5A instead of 6. The latest version of the product used in this paper is 5A (downloaded February 2018). The references to the product documentation will be added in the paper.

**7)** 2.3 Where does the information on bright-band height and width come from? Also precipitation type and rain indicators? Please add.

The parameters are extracted from the TRMM 2A23/2A25 and GPM 2AKu products themselves. In the paper we refer to Table 3 of Warren et al. (2018) for the exact list of parameters. We will also clarify that in the text as:"*Several meta-data parameters were extracted from the TRMM 2A23 and GPM 2AKu products for each SR gate, such as [...]*"

The following table describes which parameters were extracted for each product, and how they are used in the analysis. This table will be added to the documentation in the code repository, and also to a supplementary to the paper.

Table A. Parameters extracted from TRMM 2A23 and GPM 2AKu products and the derived variables used in 3D matching

| 2A23 (TRMM) | 2AKU (GPM) | Derived variable |
| --- | --- | --- |
| rainFlag | flagPrecip | Rain/no-rain indicator |
| rainType | typePrecip | Precipitation type |
| status | landSurfaceType | Surface type |
| HBB | heightBB | Brightband height |
| BBwidth | widthBB | Brightband width |
| dataQuality | dataQuality qualityBB qualityTypePrecip | Overall data quality |
| correctZFactor | zFactorCorrected | Attenuation-corrected reflectivity |
| sclocalZenith | localZenithAngle | Zenith angle |
| - | binClutterFreeBottom | Range bin number for clutter free bottom |

**8)** 2.3 GR data are acquired every 9 minutes, but matched within a 5-min window. How is this done?

With GR sweeps being repeated every 9 minutes, the maximum time difference between overpass and the closest GR sweep would be 4.5 minutes. With a buffer of 30 seconds, we set five minutes as the search window *before* and *after* the overpass. We will clarify that issue in the paper.

**9)** Figures 5-7. Sub-plot (e) is a great way to visualize this kind of result, but clearer colours are needed. I'm suspecting that light-gray points are covered by dark grey points. A colour table might be a better approach, perhaps combined with slightly smaller point sizes.

We agree that the very light colors for small quality values are difficult to interpret. Then again, we were, after some experiments, unable to adequately convey the visual message of weighting low quality samples less than high quality samples by using two different colors at both ends of the colormap. A color lookup table could not resolve the issue. Instead, we decided to start the "left" end of the colormap with a darker color, so that low quality samples become more visible. We will also implement the referee comment to decrease the point size in order to minimize overlaps.

**10)** Figure 5 caption: Replace ZPR with ZSR
$Z_{PR}$ will be replaced with $Z_{SR}$ in the caption

**11)** Figure 8. Might want to clarify in the caption that data from Jan-Apr are not used because this is the dry season.

The suggestion will be implemented.

**12)** Just a thought: what impact can radome wetting have on the results? Radome wetting is still an issue even if you exclude data near the radar. But is it an issue at all at S band?

In their review paper on sources of uncertainty, Villarini and Krajewski (2010) quoted Austin (1987) in that *"the radome attenuation is significant only for wavelengths smaller than or equal to 5 cm and negligible for wavelengths as long as 10 cm"* (i.e. S-band). Merceret and Ward (2000) reported wet radome attenuation for S-band to remain below 1 dB for rainfall intensities up to 100 mm/h. In summary, we expect wet radome attenuation to be a negligible effect in our study, and we will reference Austin (1987), Merceret and Ward (2002), and Villarini and Krajewski (2010) in order to support that assumption.

**Referee Comment # 1 (by Marco Gabella)**

[…]

**0)** I would also suggest the make the final part of the title more specific, for instance " … by using a quality filter based on beam blockage fraction"
The title will be updated to "Enhancing the consistency of spaceborne and ground-based radar comparisons by using beam blockage fraction as a quality filter".

**1)** I hope the authors can agree with the following three considerations:
    a) Quantitative interpretation of radar measurements are based on A MODEL of the
    backscattering targets.
    b) Such A MODEL is an approximation of a very complex reality (Nature).
    c) There is never sufficient information in radar measurements to resolve such complexity.

Having said that, I think I can now recommend more emphasis in the text related to the very different wavelengths and sampling volumes (for instance, you may want to have a look at the figures in the paper by Joss et al., 2006) characterizing GR (3 GHz) vs SPR (14 GHz, attenuating frequency!). Yes, one can try to correct for attenuation (e.g., Iguchi et al., 2000), he can even try to convert Z from 10 to 2 cm, but the uncertainties affecting the retrieved quantities are large! (See a) b) c) above … )

By the way, when introducing Eq. (2), you mention Cao et al. (2013) and coefficients in Table 1 for dry snow and hail … I have just quickly opened the pdf and saw that Table 1 lists (retrieval/ simulated) BIAS and RMSE?!?

I am confident that after (re-)considering the above mentioned issues, after thinking of the (necessarily) simplifying approach for beam occultation correction1 (Gaussian shape for the main lobe of the antenna radiation pattern, instead of the simple and practical linear approach proposed by Bech et al., which is an unrealistic "top-hat" radiation pattern), …, the authors will feel more comfortable with what they call "short term variability" at page 15, line 7; furthermore, they will not list "short term variability" at the first place, rather … at the last one!

First, we would like to thank the referee for spotting the mix-up in references: In fact, the conversion coefficients were reported in another paper of Cao et al. (2013), and we will correct the reference accordingly:

Qing Cao, Yang Hong, Youcun Qi, Yixin Wen, Jian Zhang, Jonathan J. Gourley, Liang Liao (2013): Empirical conversion of the vertical profile of reflectivity from Ku-band to S-band frequency, J. Geophys. Res. Atm., 118, 1814-1825, doi:10.1002/jgrd.50138.

Second, we appreciate very much that the referee puts our discussion of "short term variability" into perspective. In order to avoid misunderstandings, though, we would like to emphasize that our notion of "short term variability" does *not* necessarily imply short term variability of ground radar (mis-)calibration. In the paragraph on "short term variability", we reiterate the obvious result that our bias estimates vary substantially between overpasses. We then enumerate potential causes for that variability. Admittedly, putting "hardware instability" first in that enumeration might not be justified if we intended to list the causes ordered by their relevance to explain variability. However, the present study does not provide the scope to further disentangle and rank the different sources of uncertainty, or, in other words, the different sources of $\Delta Z$-variability between overpasses. Yet, we agree with the referee that the role of different wavelengths and sampling volumes, the role of attenuation correction at Ku band, the limitations of assuming a Gaussian antenna pattern, as well as the general limitation of the underlying backscattering model have not been sufficiently highlighted in our list of potential causes of variability. We will add the corresponding discussion and related references to the paragraph.

We will also implement the referee's suggestion to revise the order of points discussed on page 15 of the original manuscript. We will also expand the label of the paragraph from "short term variability" to "Short term variability of bias estimates between overpasses" in order to avoid misunderstandings. In line with comment #15 of this referee, the order will be changed to: 1) Effect of quality weighting on bias estimation, 2) GPM and TRMM radars are consistent, 3) Change of bias over time, 4) Short term variability of bias estimates between overpasses.

**2)** What happened to the GR in 2014? (lines 20-22, page 15 and Figure 8): +1.4 dB overestimation, after two year of clear and "heavy" under-estimation! (–4.1 dB and –2.5 dB, respectively). What a jump! Was it hardware related? Software? Both? From a weather service viewpoint, it is interesting that this paper bring in the important concept of GR calibration and monitoring, see e.g. the recent successful workshop (https://www.dwd.de/EN/specialusers/research_education/seminar/2017/wxrcalmon2017/wxrcalmon_en_node.html) However, if the authors provided possible explanations of what happened, the paper would become even more interesting and valuable. If you are interested in knowing more regarding monitoring and calibration of modern radar, you may find recent paper regarding: the Transmitter chain (e.g., Reimann et al., 2016); the Receiver chain (for instance, using the Sun: Gabella et al., 2016; Hubbert, 2017,) both Transmitter and Receiver chains, using a 24 GHz vertically pointing radar and disdrometers (Frech et al., 2017).

- Gabella, M.; Boscacci, M.; Sartori, M.; Germann, U. Calibration accuracy of the dual-polarization receivers of the C-band Swiss weather radar network. Atmosphere, 2016, 7, 76.
- Reimann, J.; Hagen, M. Antenna pattern measurements of weather radars using the Sun and a point source. J. Atmos. Ocean. Technol. 2016, 33, 891–898.
- Frech, M.; Hagen, M.; Mammen, T. Monitoring the absolute calibration of a polarimetric weather radar. J. Atmos. Ocean. Technol. 2017, 34, 599–615.

However, maybe, using a more robust definition for the SPR-GR reflectivity Bias, it will come out that the jump is smaller than 3.9 dB; for what concerns the assessment of the Mean Field Bias and its statistical evaluation, please have a look at following point #3)

As of today, we were unable, despite repeated attempts, to retrieve detailed information on maintenance operations from the radar operator. The only information provided by PAGASA engineers was that hardware replacements happened in 2014 and 2015, and the supplier changed in 2016, as already elaborated in our response to Short Comment #1 (comment no 3). As much as we appreciate the recommendations and background information on transmitter and receiver chains provided by the referee, we hope he agrees that, based on the level of information provided by the operator, any discussion of specific causes for the jumps would remain speculatory. We agree with the referee that this lack of information - that should exist  somewhere - leaves us somehow dissatisfied.

**3)** From my viewpoint, the study is a bit limited in the definition of Bias assessment and the corresponding statistical metrics for the evaluation. For instance, it is going to be straightforward for the authors to derive other statistical parameters and present them in a summarizing Table that can complement the nice and informative figure 8. First of all, in addition to the annual mean of {$\Delta$ZdB*} (lines 20-22, page 15) also the standard deviation of {$\Delta$ZdB*}. Then, I would suggest a more robust definition for the Bias: instead of using dBZ, you use Z values in linear units: Z=10(dBZ/10). Then you derive a weighted average for the numerator (denominator) using linear Z of the GR (SPR). Finally, you compute 10 Log of such ratio (dB). This annual Log_of_the_MFB is more resilient than the Mean_of_the_Log presented in the paper. To avoid weighted-average or in a probability matching scheme, you may want to consider only bins with QBBF larger than say, 0.9 (or larger). After having done this selection, consider the difference (BIASxx) between different quantiles (probability matching): xx=50

(median), 75, 84, 90, 95, 99 percentiles. Maybe, BIASxx is not constant, rather it depends on the percentile? Finally, for these QBBF-selected bins, you may explore the value of the average bias E{ΔZdB} as a function of the intensity of the echo of the GR (using for instance intervals of 3 or 5 dBZ; obviously, you will have less and less samples for larger values of dBZGR). Does this Mean_of_the_Log Bias remain more or less constant? Or do you see a trend? (Maybe, SPR has residual attenuation for large reflectivity values?). Interesting, is not it?

We would like to thank the referee for sharing these ideas. We will implement the suggestions as follows:
- We will add a visual representation of the standard deviation of the annual mean {ΔZdB*} in figure 8;
- We will recompute the bias estimates based on the referee's suggestion to first convert reflectivity to linear units before computing the weighted average;
- We will analyse the sensitivity of results in case we replace the weighted average by a simple quality threshold below which the samples will be discarded in the computation of calibration bias; however, we have the feeling that the paper is already very long, so we suggest to put the results of that analysis in a supplementary and only briefly refer to that in the main paper. Of course, using only partial beam blocking as a quality variable has very specific implications as to the effect of thresholding: any additional sample that exhibits a higher degree of partial beam blockage, and that we include in our computation of average reflectivity, will lead to a lower estimate of average ground radar reflectivity in the sample. Then again, reducing sample size through excessive filtering increases the standard deviation. That problem cannot really be resolved, but using the weighted average appears to us as the least arbitrary solution;
- Finally, we will add an analysis in which we investigate the dependency of our bias estimate on the intensity of the ground radar echo. Again, we suggest to present the results of that analysis in a supplementary, and only briefly discuss them in the main text.

**4)** Finally, the last issue is related to literature: while several TRMM PR vs GR papers are listed, there is a lack of DPR-related studies and DPR technical literature. regarding the latter, I have suggested at the end (GPM related references), three papers published in 2014 and 2015. Regarding the former, I have listed our recent DPR-related studies in the complex terrain of Switzerland; I am confident the authors will be able to find additional GPM papers also in other parts of the world. Furthermore, is Cao et al. double citation (at page 6) correct? Does Morris and Schwaller (2009) exist? (line 7, page 1 citation) Regarding GPM, please, do not forget to mention that your analysis neglect Ka-band observations (please briefly discuss the reason of such a choice).

Indeed the references were incomplete, missing DPR technical literature. References will be added accordingly, including intercomparisons between ground radar and the GPM DPR.
We will remove one of the references to Cao et al. (2013) on page 6, and correct the actual reference (as already pointed out above). The Morris and Schwaller (2009) citation mistake will also be corrected.

We will add a brief note that GPM Ka-band observations have not been considered in the present study, reasons being Ka band being more prone to attenuation, and limited validity of the Rayleigh scattering hypothesis in a substantial portion of rainfall cases (see e.g. Baldini et al. 2012).

[...]

**5)** Introduction
- Line 4: … to monitor the bias of the gauge adjustment factor to be applied to precipitation estimates of the GR.
- Lines 6: … to quantify the GR reflectivity bias with respect to the reference (namely, SR reflectivity value after conversion from Ku-band to S-band).

The above mentioned lines in the abstract will be revised accordingly.

**6)** In fact, I would propose the following terminology:
- Setting the Bias as close as possible to 0 dB between radar QPE and in situ measurements: *gauge-adjustment*
- Assessing the Bias between reflectivity of two radars: *relative calibration*
- Forcing to 0 dB the Bias in measured Power (dBm) between an external or internal reference Noise Source and the radar at hand: *absolute calibration*

We agree with the suggestion and will revise the manuscript accordingly, introducing the labels "gauge adjustment", "relative calibration", and "absolute calibration" in the first section.

**7)** Section 2.1.2
The fact that only 283 overpasses were within the selected, reasonable 120 km range should be mentioned here. There is no reason to wait until former(see **) sec. 3.1.1. Similarly, you can at least anticipate that the number will considerably decrease upon conditional requirements such as min. # of "wet" pixels, time difference, min. # of bins above both GR and SPR sensitivity.

We agree that it might be confusing to state numbers of overpasses or valid samples without already anticipating the effect of spatial limitations or additional filter requirements. We will revise the manuscript accordingly by stating these effects early in the paper.

**8)** Question: have you only used only months from June to November? Not clear from the text. Please rephrase. In fig. 8, I see two overpasses in December (2012 and 2014). By the way, in Dec. 2012 E{$\Delta$ZdB*} is almost 5! dB, while the annual average is -4.1 dB?!? (see my previous points 2) and 3) )

We used only the months from June to December, which coincides with the rainy season in the area. We will clarify that in the text. As for the case of December 2012, upon checking the particular GR-SR match (December 5, 2012), the value of $\Delta$Z* is indeed very high (4.2 dB) compared to the average. Looking at the GR and SR data, the number of samples seems sufficient (n=382), and the GR overestimation is

consistent for the different elevation angles. As a result, we cannot provide a consistent explanation for this outlier.

**9)** By the way, I would propose the following structure for Sections and Subsections
2. Data
2.1 Spaceborne Precipitation Radar (SPR)
2.2 GR

3. Method
3.1 Partial beam shielding and quality index based on beam blockage fraction
3.2 SPR-GR volume matching
3.3 Assessment of the average reflectivity Bias

Section 3.1:  I would move it (including former fig. 4) inside the new Section 3.1 (former Sec. 2.2)

4. Results and Discussion
4.1 Single event comparison
4.1.1 Case1
4.1.2 Case2
4.2 Overall June-November comparison during the 5-year observation period

We would like to thank the referee for this suggestion. We agree with the proposed structure, and will update the paper accordingly.

**10)** Page 5, Line 4-5: Please delete the sentence, the reader is able to read the simple algebra in eq. (1). The sentence has been deleted.

**11)** Page 10, Lines 19-25: misleading. I cannot possibly agree. On the contrary, my interpretation is that partial beam blockage plays approximately the same role (0.7 dB difference between the silly estimate that include blockage and the conservative one that exclude all cases where BBF > 0.5). Please rephrase.

We agree that our interpretation was hard to follow, and the corresponding paragraph kind of confusing. That was also pointed out in comment #11 of referee #2. We will rephrase that part of the paper, and hope that our point will become clearer - because we still think it is quite an important one! At a low elevation angle, substantial parts of the sweep are affected by **total** beam blockage. The affected bins are either below the detection limit, or they do not exceed the GR threshold specified in Table 2 of the manuscript. As a consequence, these bins will not be considered in our matched samples, and will thus not influence our bias estimate - irrespective of using partial beam blockage as a quality filter. At a higher elevation angle, though, the same bins might not be affected by **total** beam blockage, but by **partial** beam blockage, as also becomes obvious from Fig. 4 of the manuscript. If we consider these bins in our matched samples, they will cause a systematic error in our estimate of calibration bias,

unless we use the partial beam blockage fraction as a quality filter by computing a quality-weighted average of reflectivity. As a consequence, the effect of quality-weighted averaging (with partial beam blockage fraction as a quality variable) can be most pronounced at "intermediate" elevation angles, depending of course on the specific topography and the relative position of the ground radar. We had referred to that effect as "counterintuitive" since one might naively expect that the detrimental effects of beam blockage on our estimate of calibration bias would *generally* decrease with increasing elevation angle.

**12)** Page 13: Would you please add a complementary figure at ELEV= 1.5 for the 1.10.2015 overpass? Just like you did for the 8.11.2013 overpass.

We thank the referee for the suggestion, however, we are hesitant to add the additional figure as it does not provide additional insight as compared to the comparison of two sweeps for 2013-11-08, while adding to the length of the manuscript. As a compromise, we suggest to add the additional figure to the supplementary material.

**13)** Page 14 and Figure 8. Some journals ask for a graphical abstract as a self-explanatory image to appear alongside with the abstract. I think Fig. 8 would be perfect for such scope. It is nice and rich of information. Suggestion: could you please use color. For instance, the 1.10.2015 and 8.11.2013 overpasses could be in color. By the way, the 8.11 circle in picture a) seems to be very close to 0 dB, while in Fig. 7 it is written that $E\{\Delta ZdB*\}$ is -1.1 dB. Am I missing something? Is it related to what you wrote in lines 3-6? These sentences are not clear to me, could you rephrase, please? Furthermore, regarding picture b), do not forget to emphasize that if the QBFF works properly then: $E\{\Delta ZdB*\}- E\{\Delta ZdB\}$ should be negative in 2012 and 2013 (almost all the point in a) are below the 0 dB dotted line), positive in 2014 (almost all the point in a) above the 0 dB dotted line …).

We thank the referee for the suggestion to highlight the two case studies in Figure 8 by color, and we will implement the suggestion accordingly. We are also grateful for suggesting a potential error, however, in this case, we do not agree: the triangle for Nov 8, 2013, represents correctly the bias estimate on that date, as an average over samples from all sweeps (-3.7dB). Apart from that, Fig. 7 refers to the overpass on October 1, 2015.

We also thank the referee for pointing out the issue of negative differences $E\{\Delta Z*\}$ - $E\{\Delta Z\}$ in Fig. 8b which we missed to discuss sufficiently in the manuscript. First, we would like to clarify that if the QBBF works properly, the difference $E\{\Delta Z*\}$ - $E\{\Delta Z\}$ should be positive - the areas suffering from partial beam blockage registers weaker signals (i.e. lower reflectivity) than expected producing the "old" lower mean bias, and giving them low weights in the calculation of mean bias brings the "new" (quality-weighted) mean bias up. In the same vein, the difference in standard deviation should be negative - the "new" standard deviation that considers quality is lower than the "old" standard deviation that does not consider quality, so that the difference between "new" and "old" standard deviation is negative. The negative differences $E\{\Delta Z*\}$ - $E\{\Delta Z\}$ are therefore inconsistencies, caused by the effect of filtering in the case of very small sample sizes. We will include this clarification in the revision.

**14)** Page 15. I would change the order of your points and list your point (1) at the end, as # (4) [see my comment 1) at page 1)]. I would start from (3), which is the scope of this paper: indeed an intelligent weighted-average based on QBFF shows a better standard deviation of ΔZdB*. By the way, I recommend adding a table and/or a figure (histogram) that summarizes the statistical properties of $\sigma*\{\Delta ZdB*\}$ and $\sigma\{\Delta ZdB\}$. Then, I would introduce the important result regarding the consistency of GPM and TRMM radars, followed by the changes of the bias in time

As already pointed out in our response to comment #1 of the referee, we will change the order of points as suggested. However, we decided not to introduce additional figures in terms of histograms of bias, differences in bias, or standard deviations. These histograms would have to be provided separately for each year, because it is obvious from the time series that they would represent different populations. Apart from avoiding to introduce many new figures, the informative value of these histograms is not too high due to the limited number of samples. Instead, we will implement the referee's suggestion from comment #3 by including the standard deviation of the annual mean {ΔZdB*} in Fig. 8a.

**15)** Page 16.
Line 5, delete coherent.
The word "coherent" has been deleted.

**16)** Line 14-16. Sorry, you cannot summarize the (mis-) calibration of the GR by simply going from 2012 (–4.1 dB) to 2016 (+0.6 dB) and omit, for instance, the +1.4 jump in 2014. [see my comment 2) at page 2)].

We will revise the manuscript accordingly by providing a more complete and coherent summary of the temporal changes of our bias estimate.

**17)** Line 17-19. Pleonastic. I would delete it.

We would like to refer to our response to the referee's comment #11: we hope that we were able to clarify a misunderstanding there. Given that the referee agrees with our clarification, we think that lines 17-19 on page 16 are not pleonastic, but rather an important note to emphasize that moving to higher elevation angles does not necessarily help to avoid the problems introduced by beam blockage in the specific case of comparing GR and SR observations. Nevertheless, we will also revise the corresponding paragraph in the conclusions section in order to make it more comprehensible.

**18)** Line 26. Why do you discuss C-band radar technology ?

Lines 24-28 on page 16 of the original manuscript were intended to provide a brief perspective for future studies, in which we mention that for C-band radars, it would be important to include path-integrated attenuation as a quality variable. In the revised version, we will clarify that point.

**19)** Minor points

My proposal for radar acronyms: 2-character for ground, namely GR; 3-character for satellite radar. Would you please use TPR for TRMM, DPR for GPM and SPR in those cases where you refer to both, independently of the platform

We appreciate the suggestion. Yet, we think that distinguishing the different spaceborne platforms via acronyms might cause more confusion than clarification, in particular since we rarely address the different platforms separately in the main text. We would thus prefer to stick with GR vs. SR in general.

**Referee Comment #2 (Anonymous)**

[...]

**Comments on other sources of uncertainty in calibration assessment:**

**1)** Attenuation at Ku-band:
The authors should address the uncertainties with attenuation correction at Ku-band. The attenuation correction tech. used for just Ku-band is the HB-SRT method (Seto and Iguchi 2015). It is known that using the HB method alone does not work well in higher rain rates (> 20 mm hr-1, Seto and Iguci 2011, but as low as 12 mm hr-1 Rose and Chandrasekar 2005). Furthermore, the SRT method is more uncertain over land (larger standard deviation of the surface backscatter cross-section, Meneghini et al. 2000). It is anticipated that since the radar is located in the tropics both of the issues above could occur (more likely in convective precipitation). Please discuss these uncertainties and how they could impact your results of the bias correction.
It is mentioned in the conclusions that for C-band attenuation correction is vital, but GPM and TRMM are Ku-band, thus isn't it vital as well?

We agree that attenuation correction is vital for both GPM and TRMM at Ku-band, and there is certainly a large body of literature concerned with the related effects, including the effects of nonuniform beam filling (NUBF) on the attenuation correction procedure. In the present study, we have only used the attenuation-corrected reflectivity values without considering the uncertainty associated with the correction procedure. In the revised manuscript, we will explicitly refer to the uncertainty introduced by attenuation correction. We will also, in the conclusions, provide an outlook on including the spaceborne reflectivity observations in the framework of quality-weighted averaging, just as we suggested for the ground radar observations. That would imply to use the estimates of PIA which are provided through the SR meta-data as a quality variable and thus to consider it in the quality-weighted average of SR reflectivity in the matched samples.

**2)** Ground Clutter for the SR:
In radar gates near the surface, with respect to the SR, ground clutter is a problem. How are the authors dealing with ground clutter from the SR? Are they using gates below the lowest clutter free bin estimate (included in the GPM file)? If so, is the lowest clutter free gate being

assigned to all the gate below it? If you plot it out, a lot of times that's what is done. Essentially the data looks smeared from the lowest clutter free bin to the surface, which isn't to realistic and it is suggested to just not consider these gates. Please comment on this, potentially in Section 2.3. If you are including these interpolations, you may wish to not (it will introduce error).

Thanks for pointing out this issue which has not yet been sufficiently clarified in the original manuscript. While TRMM 2A25 contains a clutter flag for the variable "Corrected Z-factor" (-8888 indicates ground clutter), the GPM 2AKu product contains a variable "binClutterFreeBottom" to indicate the lowest clutter free bin in a ray. In both cases, TRMM and GPM, we use the SR clutter information to discard the affected bins. We will clarify that point in the revised manuscript, using both table 2 (filtering criteria), and the new table with metadata variables that we introduced as a response to comment #7 of SC1 (as part of the the supplementary).

**3)** NUBF:
Please also include some discussion of the potential impacts of non-uniform beam filling (NUBF) on your analysis. Edges of large systems, individual cumulus showers could result in NUBF in SR because of the quasi-large footprint. Lowering the reflectivity value in the gate.

We agree that non-uniform beam filling can cause errors in particular for the SR platform which might become more pronounced in case of path-integrated attenuation is present and being corrected for. Durden et al. (1998) provided an excellent discussion of potential effects. Han et al. (2018) attempted to consider the effect in case GR and SR observations are matched, by using the - comparatively highly resolved - GR observations in order to compute the standard deviation of reflectivity in an SR footprint as a measure of NUBF. From the literature, it is hard to tell how much systematic error is introduced in SR measurements by the effects of NUBF. However, the three comments of this referee (reg. attenuation, clutter, NUBF) were very helpful for us to understand the necessity of extending the framework of quality-weighted averaging to the SR, too. So while we consider our present manuscript as a proof-of-concept in the consideration of quality, follow up studies should attempt to achieve a more general implementation that not only includes additional quality variables for the GR data, but that also applies these to the SR observation which already come with extremely rich and helpful meta-data to support such attempts. While our study tries to minimize the effects of NUBF (by setting a minimum fraction of GR bins within the SR footprint to exceed a minimum reflectivity threshold, see table 2 of the original manuscript), a future framework for SR quality might rather consider the variability of GR bins in the SR footprint, as suggested by Han et al. (2018).

**Specific Comments:**

**4)** Page 2, line 5: Please add the Kummerow et al. (1998) paper for TRMM, and the Hou et al. (2014) for the GPM reference (page 2, line 6). This will help readers who are not entirely familiar with both platforms.

The Kummerow et al. (1998) and Hou et al. (2014) citations and references have been added.

**5)** Page 6, line 3: "The gates below and above the brightband were considered in the comparison". Please provide a brief reason why this is done. I do not want to assume the author's reasoning.

*According to Warren et al. (2018), the frequency-corrected reflectivities within the melting layer (bright band) appear underestimated compared to the ones below and above the melting layer. In addition, while usually the samples above the brightband are used in GPM validation, there are significantly more samples below the melting layer, especially in a tropical environment such as the Philippines.*

**6)** Figure 4 & Section 3.1: It is not clear what you are plotting. The figure titles state the quality index but the figure caption and text states beam blockage fraction. Please clarify.

*The caption has been updated to match the figures: Quality index map of the beam blockage fraction for the Subic radar at (a) 0.0° (b) 0.5° (c) 1.0° and (d) ° elevation angles.*

**7)** Section 3.1.1: Why are the number of overpasses here different than when they were listed earlier (section 2.1.2)? I am referring to the numbers before applying the criteria in Table 2.

*Applying the criterion of "Minimum number of pixels tagged as rain = 100" eliminates several overpasses. Only this criteria affects the number of overpasses, not the others listed in Table 2. We will clarify this in the paper.*

**8)** Case studies (Section 3.1.2 and 3.1.3): Could you include the mean BB level height? You can add it to the bottom right with the other statistics. Also comment on fraction of stratiform vs convective. These two will help readers assess the amount of attenuation and NUBF that could be involved (e.g. uncertainty in the SR measurements).

*The mean BB level height will be added to the figure as suggested. While stratiform rain dominates the precipitation type for most cases, convective rain is significantly represented, hence we decided to keep both rain types in the analysis.*

**9)** Figure 5 + 6 + 7 a and b: Suggestion. Consider changing the colorscale to one that is perceptually uniform and color-deficient friendly. For example, try the HomeyerRainbow or the LangRainbow included in Pyart (https://github.com/ARM-DOE/pyart)

*We thank the referee for the suggestion. Upon trying the different colormaps proposed, we decided that we will go with the HomeyerRainbow colormap. The figures will be updated to reflect the new colormap.*

**10)** Page 10, Line 12: "Major parts of that sector did not receive any signal due to total beam blockage". Where is this occurring? The reader can refer back to Figure 4, but it might be

helpful to outline the circles with a thin black line in Figure 5d where there is SR data, but no GR data. That way the readers would see where there is 100% beam blockage and thus no signal from the GR, but also gain insight of size of the precipitating system.

The figures for the case studies show only the matched bins, but the referee is right, information such as location of bins where there is SR signal but no GR signal and the size of precipitating system are not conveyed. We will address this by showing all the available SR bins for the first panel and outlining the circles with SR data but no GR data in black, as suggested.

**11)** Page 10, Line 24-25: "That might be considered counterintuitive, as one might expect the blockage to disappear with higher elevations". Please provide some discussion explaining why this is the case.

We thank the referee for pointing out the lack of adequate explanation. As can be seen also from the comments of referee #1, this paragraph appears to be confusing in the original manuscript. We will revise the paragraph accordingly in order to make our point clearer. Please also refer to our response to the comment #11 of referee #1.

**12)** Page 16, Lines 13 – 16. 'We could' and 'we could also' imply that you did not conduct this analysis when it seems you have. I suggest to change these phrases to be definitive. 'We showed that…' 'we also demonstrated that…'
The sentences will be updated as suggested.

**Technical corrections:**

**13)** Page 3, line 20: The most current GPM version is version 5, version 6 is not released yet.
The version will be corrected (version 5A instead of 6).

**14)** Page 18, line 18: Reference Cao et al. 2013 is incorrect. It should be:
Empirical conversion of the vertical profile of reflectivity from Ku-band to S-band frequency
We apologize for the mixup. The citation and reference will be corrected to refer to
Cao, Qing, Yang Hong, Youcun Qi, Yixin Wen, Jian Zhang, Jonathan J. Gourley, and Liang Liao. 2013.
"Empirical Conversion of the Vertical Profile of Reflectivity from Ku-Band to S-Band Frequency." *Journal of Geophysical Research: Atmospheres* 118 (4): 1814–25. https://doi.org/10.1002/jgrd.50138.

**15)** The reference Warren et al. should be 2018, published Feb 2018 in J. Atmo. + Ocean. Tech.. Page 2, line 8;Page 3,line 25;Page 5,line 11;Page 15,line 14
The citations and reference will be corrected.

**16)** Figure 4: Missing y-ticks and tick labels on bottom left subplot

Axis labels will be restored in Figure 4. The color scheme has been changed so that the lightest color is made a bit darker for better visibility in Figures 5-7 subplots d and e, following the suggestion of another reviewer.

**17)** Page 8, line 5-6. No need for new paragraph. You can combine the two.
The paragraphs will be combined as suggested.

**18)** Figure 5: Figure caption has Zpr instead of Zsr
$Z_{PR}$ will be replaced with $Z_{SR}$ in the caption

**References**

Austin, P.M. (1987): Relation between measured radar reflectivity and surface rainfall. Mon Weather Rev., 115, 1053-1071.

Baldini, L., V. Chandrasekar, D. Moisseev (2012): Microwave radar signatures of precipitation from S band to Ka band: application to GPM mission, European Journal of Remote Sensing, 45:1, 75-88, DOI: 10.5721/EuJRS20124508.

Biswas, S. K. (2017): Cross Validation of Observations from GPM Dual-Frequency Precipitation Radar with S-Band Ground Radar Measurements over the Dallas — Fort Worth Region. In *2017 IEEE International Geoscience and Remote Sensing Symposium (IGARSS)*. Fort Worth, TX, USA: IEEE. https://doi.org/10.1109/IGARSS.2017.8127393.

Cremonini, R., Moisseev, D., and Chandrasekar, V. (2016): Airborne laser scan data: a valuable tool with which to infer weather radar partial beam blockage in urban environments, Atmos. Meas. Tech., 9, 5063-5075.

Durden SL, Haddad ZS, Kitiyakara A, Li FK (1998): Effects of nonuniform beam filling on rainfall retrieval for the TRMM precipitation radar. J. Atmos. Oceanic Technol. 15: 635.

Han, J., Z. Chu, Z. Wang, D. Xu, N. Li, L. Kou, F. Xu, Y. Zhu (2018): The establishment of optimal ground-based radar datasets by comparison and correlation analyses with space-borne radar data, Meteorol. Appl. 25, 161-170.

Kucera, P. A., W. F. Krajewski, and C. B. Young (2004): Radar Beam Occultation Studies Using GIS and DEM Technology: An Example Study of Guam. Journal of Atmospheric and Oceanic Technology 21 (7): 995–1006.

Merceret, F. J., J. G. Ward (2002): Attenuation of Weather Radar Signals Due to Wetting of the Radome by Rainwater or Incomplete Filling of the Beam Volume, Technical Report NASA/TM-2002-211171, NAS 1.15:211171, 20 p., URL: https://ntrs.nasa.gov/search.jsp?R=20020043890

Villarini, G., W. F. Krajewski (2010): Review of the Different Sources of Uncertainty in Single Polarization Radar-Based Estimates of Rainfall, Surv Geophys (2010) 31:107–129

---

## Author Comment (AC3) · 29 Jun 2018

The comment was uploaded in the form of a supplement:
https://www.atmos-meas-tech-discuss.net/amt-2018-101/amt-2018-101-AC3-
supplement.pdf

---

## Author Response (AR1)

**Manuscript revision and response to the referees**

We would like to thank the referees again for their valuable comments. With this document, we provide a comprehensive response that attempts to track the discussion from the original referee comments over our response in the interactive discussion to our final response including the decision on specific changes in the manuscript. To allow for this tracking, we use the following color code:

black: original referee comment
blue: our original response in the Interactive discussion
green: our final response and the specific changes made in the manuscript revision

We are confident that addressing the referee comments has substantially improved the paper, and we hope that the quality of the paper now allows for publication in AMT.

**Short Comment # 1 (by Daniel Michelson)**

[…]

**1)** How to characterize data quality? Significant effort has been devoted to systematic representation of weather radar data quality in Europe, through COST Actions (717, 731) and EUMETNET OPERA, giving framework approaches for dealing with data quality. The authors have followed Zhang et al. (2011) for representing data quality resulting from beam blockage. It would be useful to have some context in the paper acknowledging previous work and including a rationale for choosing the Zhang et al. approach.

**RESPONSE:** We now refer to the framework for quality representation in OPERA, and briefly compare it to the approach of Zhang et al. (2011). It should be noted, though, that our general approach is open to other definitions of overall data quality, as long as such an overall value can be used to compute weights. Furthermore, we used only one single quality variable (beam blockage fraction). Hence, the challenge to aggregate several variables to a single index is not prominent in our study.

**ACTION:**
The following paragraph was added to the beam blockage subsection:
*An alternative function to transform partial beam blockage to a quality index has been presented in other studies (Figueras i Ventura and Tabary, 2013; Fornasiero et al., 2005; Osródka et al., 2014; Rinollo et al., 2013) where the quality is zero (0) if BBF is above a certain threshold, and then linearly increases to one (1) above that threshold. It should be noted that these approaches are equally valid and can be used in determining the quality index based on beam blockage.*

The following sentence has also been added to the conclusions:
*In addition, with the significant effort devoted to weather radar data quality characterization in Europe (Michelson et al., 2005), and the number of approaches in determining an overall quality index based on*

*different quality factors (Einfalt et al., 2010), it is straightforward to extend the approach beyond beam blockage fraction.*

**2)** What advantages does this work offer when it comes to addressing topographical beam blockage with GR compared to previous work? The paper references Bech et al. (2003) which is a benchmark paper. There are other implementations of the same approach that use other DEM data, e.g. GTOPO30. The authors' use of high-resolution SRTM data is interesting, but are the results better than using ~1 km GTOPO30 data?

**RESPONSE:** Generally, any increase in DEM resolution should be expected to increase the accuracy of our estimate of the beam blockage fraction (see e.g. Kucera et al. 2004). That effect could be particularly prominent in the near range of the radar as has been shown e.g. for high-resolution airborne laser DEMs by Cremonini et al. (2016). Yet, there is no reference (truth) that could be used to actually verify beam blockage estimates for any underlying DEM. What could be done, however, is to repeat the analysis with GTOPO30 data (i.e. 1 km resolution), in order to investigate the sensitivity of results, or whether our estimate of calibration bias becomes more or less consistent using GTOPO30. Yet, we already consider the paper as quite long, especially considering the requested changes in the course of this review process. We are thus hesitant to include such additional analysis that is not expected to add substantial new insight. Instead, we will add a brief discussion on the potential effects of DEM resolution on the quantification of partial beam blockage.

**ACTION**: We added the following sentence in section 3.1:
*"While Bech et al. (2003) used the GTOPO30 DEM at a resolution of around on kilometer, higher DEM resolutions are expected to increase the accuracy of estimates of beam blockage fraction, as shown by e.g. Kucera et al. (2004), in particular the near range of the radar (Cremonini et al., 2016)."*

**3)** GR calibration. There is one sentence in 3.2(2) indirectly indicating that the Subic radar may have been calibrated during the time period covered by the study. This needs to be clarified. Was the radar calibrated during this time? More than once? Are the results available? Any other maintenance that could have impacted calibration levels? This is very important to understand results like those presented in Figure 8. Also, the methods presented in this paper have been applied to data from one GR, yet they would be much more valuable if also applied to data from a second GR. Doing so would reveal which radar is "hot" and which is "cold" and whether there are any other systematic differences that are unique to each GR.

**RESPONSE:** We entirely agree that it would be interesting to apply the methodology to another or even several other radars in order to investigate specific characteristics of individual radars, or in other words, differences and similarities. Yet, we consider this study as a proof-of-concept in which we present the underlying methodology, and show that it adds value in estimating calibration bias for a single radar. An inter-comparison with several radars would be an additional study. Such a study should not only compare the behaviour of different radars independent of each other. It should also investigate the effect of "recalibration" on the consistency between two or more radars in regions of overlap (c.f. Warren et al. 2018). However, we consider such an analysis beyond the scope of the present paper.

Unfortunately, we were unable, despite repeated attempts, to retrieve detailed information on maintenance operations from the radar operator. The only information provided by PAGASA engineers was that in 2012 and 2013, there were problems with the transmitter which caused the output power to be very low (~80 dBm as opposed to the required 89.2 dBm). The modulator was replaced in 2014. In 2015, the magnetron was replaced. In October 2016, the supplier was changed to SELEX. We also have to assume that after performing major changes in the radar hardware, the radar engineers carried out a calibration.

**ACTION:** We provided the following clarification as the last sentence of 4.2(3):

*We have to assume that a fundamental issue with regard to calibration maintenance was addressed between 2013 and 2014 from hardware changes (i.e. replacement of magnetron). Unfortunately, we were not able to retrieve detailed information on maintenance operations that might explain the changes in bias of the radar throughout the years.*

**Technical "corrections"**

**4)** Including information throughout the paper on what software calls are available and have been used is irrelevant and should be avoided. Instead, I recommend a small section following the recommendations given by Irving: https://doi.org/10.1175/BAMS-D-15-00010.1

**RESPONSE:** In the revised version of the paper, we will discuss the context of Irving (2016). In fact, our paper moves beyond the "minimum standard" suggested by Irving, since we not only provide the code (a doi pointing to the wradlib package version will be generated for the final version of the paper), but we combine the enhancement of an existing software package (wradlib, extensively documented) with a fully documented application context (a jupyter notebook), combined with the data, so that we do not have to provide a log file, as suggested by Irving, but directly allow the user to run the analysis, and modify it in order to adapt to different application contexts. However, we agree that it makes sense to remove the references to explicit function calls from the main text. In order to conform with Irving's suggestions, we also add a brief section with the description of the underlying software, its dependencies, and the notebook to reproduce the results.

**ACTION:** The subsection **2.5 Computation details** has been added in the manuscript. Explicit references to functions have been removed from the text.

**5)** References to Morris and Schwaller and Schwaller and Morris are inconsistent. In the list of references, both are given from 2011, but the paper references one from 2009. The Morris and Schwaller reference appears to be incomplete in the list of references.

**RESPONSE:** The Schwaller and Morris citations will be updated to consistently refer to Schwaller and Morris (2011). The Morris and Schwaller reference will be updated to show complete information.

**ACTION:** The citations and the reference have been updated.

**6)** 2.1.2 Version 6 of the GPM 2AKu products is stated, but is not the current version 05B? Reference(s) to product documentation are needed.

**RESPONSE:** The version will be corrected to 5A instead of 6. The latest version of the product used in this paper is 5A (downloaded February 2018). The references to the product documentation will be added in the paper.
**ACTION:** The version number has been corrected to 5A instead of 6. The reference was added in the paper.

**7)** 2.3 Where does the information on bright-band height and width come from? Also precipitation type and rain indicators? Please add.

**RESPONSE:** The parameters are extracted from the TRMM 2A23/2A25 and GPM 2AKu products themselves. In the paper we refer to Table 3 of Warren et al. (2018) for the exact list of parameters. We will also clarify that in the text as:"*Several meta-data parameters were extracted from the TRMM 2A23 and GPM 2AKu products for each SR gate, such as [...]*"

The following table describes which parameters were extracted for each product, and how they are used in the analysis. This table will be added to the documentation in the code repository, and also to a supplementary to the paper.

Table A. Parameters extracted from TRMM 2A23 and GPM 2AKu products and the derived variables used in 3D matching

| 2A23 (TRMM) | 2AKU (GPM) | Derived variable |
|---|---|---|
| rainFlag | flagPrecip | Rain/no-rain indicator |
| rainType | typePrecip | Precipitation type |
| status | landSurfaceType | Surface type |
| HBB | heightBB | Brightband height |
| BBwidth | widthBB | Brightband width |
| dataQuality | dataQuality qualityBB qualityTypePrecip | Overall data quality |
| correctZFactor | zFactorCorrected | Attenuation-corrected reflectivity |
| sclocalZenith | localZenithAngle | Zenith angle |
| - | binClutterFreeBottom | Range bin number for clutter free bottom |

**ACTION:** We have clarified the source of the brightband width and height data in Section 3.2. We have decided, however, not to introduce a supplementary section to the paper and thus we will not include

the above-mentioned Table A. Instead, we hope that the reference to Warren et al. (2018) Table 3 will suffice.

**8)** 2.3 GR data are acquired every 9 minutes, but matched within a 5-min window. How is this done?

**RESPONSE:** With GR sweeps being repeated every 9 minutes, the maximum time difference between overpass and the closest GR sweep would be 4.5 minutes. With a buffer of 30 seconds, we set five minutes as the search window *before* and *after* the overpass. We will clarify that issue in the paper.

**ACTION:** We have clarified the issue in Section 3.2.

**9)** Figures 5-7. Sub-plot (e) is a great way to visualize this kind of result, but clearer colours are needed. I'm suspecting that light-gray points are covered by dark grey points. A colour table might be a better approach, perhaps combined with slightly smaller point sizes.

**RESPONSE:** We agree that the very light colors for small quality values are difficult to interpret. Then again, we were, after some experiments, unable to adequately convey the visual message of weighting low quality samples less than high quality samples by using two different colors at both ends of the colormap. A color lookup table could not resolve the issue. Instead, we decided to start the "left" end of the colormap with a darker color, so that low quality samples become more visible. We will also implement the referee comment to decrease the point size in order to minimize overlaps.

**ACTION:** The colormap for Figures 5-7 subplot d and e have been shifted so that the left/lower end has a darker color, to increase visibility of low quality samples. The point sizes were also decreased to minimize overlaps. In addition, the colormaps of subplots a and b have been updated to the more colorblind friendly Homeyer colormap.

**10)** Figure 5 caption: Replace ZPR with ZSR
**RESPONSE:** $Z_{PR}$ will be replaced with $Z_{SR}$ in the caption
**ACTION:** Proposed changes have been implemented.

**11)** Figure 8. Might want to clarify in the caption that data from Jan-Apr are not used because this is the dry season.

**RESPONSE:** The suggestion will be implemented.
**ACTION:** Proposed changes have been implemented.

**12)** Just a thought: what impact can radome wetting have on the results? Radome wetting is still an issue even if you exclude data near the radar. But is it an issue at all at S band?

**RESPONSE:** In their review paper on sources of uncertainty, Villarini and Krajewski (2010) quoted Austin (1987) in that *"the radome attenuation is significant only for wavelengths smaller than or equal to 5 cm and negligible for wavelengths as long as 10 cm"* (i.e. S-band). Merceret and Ward (2000) reported wet

radome attenuation for S-band to remain below 1 dB for rainfall intensities up to 100 mm/h. In summary, we expect wet radome attenuation to be a negligible effect in our study, and we will reference Austin (1987), Merceret and Ward (2002), and Villarini and Krajewski (2010) in order to support that assumption.

**ACTION:**
The portion of the conclusion that briefly discusses C-band radars are updated to:
*For example, for the other C-band radars in the Philippine radar network, considering path-integrated attenuation would be vital. While we expect the wet radome attenuation to be negligible for this S-Band radar (Austin (1987), Merceret and Ward (2002), and Villarini and Krajewski (2010)), the same cannot be said for C-band radars.*

**Referee Comment # 1 (by Marco Gabella)**

[...]

**0)** I would also suggest the make the final part of the title more specific, for instance " … by using a quality filter based on beam blockage fraction"

**RESPONSE:** The title will be updated to "Enhancing the consistency of spaceborne and ground-based radar comparisons by using beam blockage fraction as a quality filter".

**ACTION:** The title has been updated to "Enhancing the consistency of spaceborne and ground-based radar comparisons by using beam blockage fraction as a quality filter"

**1)** I hope the authors can agree with the following three considerations:
      a) Quantitative interpretation of radar measurements are based on A MODEL of the backscattering targets.
      b) Such A MODEL is an approximation of a very complex reality (Nature).
      c) There is never sufficient information in radar measurements to resolve such complexity.

Having said that, I think I can now recommend more emphasis in the text related to the very different wavelengths and sampling volumes (for instance, you may want to have a look at the figures in the paper by Joss et al., 2006) characterizing GR (3 GHz) vs SPR (14 GHz, attenuating frequency!). Yes, one can try to correct for attenuation (e.g., Iguchi et al., 2000), he can even try to convert Z from 10 to 2 cm, but the uncertainties affecting the retrieved quantities are large! (See a) b) c) above … )

By the way, when introducing Eq. (2), you mention Cao et al. (2013) and coefficients in Table 1 for dry snow and hail … I have just quickly opened the pdf and saw that Table 1 lists (retrieval/ simulated) BIAS and RMSE?!?

I am confident that after (re-)considering the above mentioned issues, after thinking of the (necessarily) simplifying approach for beam occultation correction (Gaussian shape for the main lobe of the antenna radiation pattern, instead of the simple and practical linear approach proposed by Bech et al., which is an unrealistic "top-hat" radiation pattern), ..., the authors will feel more comfortable with what they call "short term variability" at page 15, line 7; furthermore, they will not list "short term variability" at the first place, rather ... at the last one!

**RESPONSE:** First, we would like to thank the referee for spotting the mix-up in references: In fact, the conversion coefficients were reported in another paper of Cao et al. (2013), and we will correct the reference accordingly:

Qing Cao, Yang Hong, Youcun Qi, Yixin Wen, Jian Zhang, Jonathan J. Gourley, Liang Liao (2013): Empirical conversion of the vertical profile of reflectivity from Ku-band to S-band frequency, J. Geophys. Res. Atm., 118, 1814-1825, doi:10.1002/jgrd.50138.

Second, we appreciate very much that the referee puts our discussion of "short term variability" into perspective. In order to avoid misunderstandings, though, we would like to emphasize that our notion of "short term variability" does *not* necessarily imply short term variability of ground radar (mis-)calibration. In the paragraph on "short term variability", we reiterate the obvious result that our bias estimates vary substantially between overpasses. We then enumerate potential causes for that variability. Admittedly, putting "hardware instability" first in that enumeration might not be justified if we intended to list the causes ordered by their relevance to explain variability. However, the present study does not provide the scope to further disentangle and rank the different sources of uncertainty, or, in other words, the different sources of $\Delta Z$-variability between overpasses. Yet, we agree with the referee that the role of different wavelengths and sampling volumes, the role of attenuation correction at Ku band, the limitations of assuming a Gaussian antenna pattern, as well as the general limitation of the underlying backscattering model have not been sufficiently highlighted in our list of potential causes of variability. We will add the corresponding discussion and related references to the paragraph.

We will also implement the referee's suggestion to revise the order of points discussed on page 15 of the original manuscript. We will also expand the label of the paragraph from "short term variability" to "Short term variability of bias estimates between overpasses" in order to avoid misunderstandings. In line with comment #15 of this referee, the order will be changed to: 1) Effect of quality weighting on bias estimation, 2) GPM and TRMM radars are consistent, 3) Change of bias over time, 4) Short term variability of bias estimates between overpasses.

**ACTION:** The mix-up in the references has been corrected. The order of the four discussed items at the end of section 4.2 has been revised according to the referee's suggestion. We enhanced the scope of the 4th item (Short term variability of bias estimates between overpasses) in order to provide more context with regard to various sources of uncertainty that still affect the comparability and consistency of GR and SR observations, and thus the stability of our bias estimates. That way, we also address comments 1-3 of Referee Comment #2. We also added a brief discussion of uncertainties of our beam blockage

estimation (e.g. Gaussian antenna pattern, beam propagation under different refraction, DEM errors and insufficient resolution, DEM interpolation errors) under paragraph 4.2(1).

**2)** What happened to the GR in 2014? (lines 20-22, page 15 and Figure 8): +1.4 dB overestimation, after two year of clear and "heavy" under-estimation! (–4.1 dB and –2.5 dB, respectively). What a jump! Was it hardware related? Software? Both? From a weather service viewpoint, it is interesting that this paper bring in the important concept of GR calibration and monitoring, see e.g. the recent successful workshop (https://www.dwd.de/EN/specialusers/research_education/seminar/2017/wxrcalmon2017/wxrcalmon_en_node.html) However, if the authors provided possible explanations of what happened, the paper would become even more interesting and valuable. If you are interested in knowing more regarding monitoring and calibration of modern radar, you may find recent paper regarding: the Transmitter chain (e.g., Reimann et al., 2016); the Receiver chain (for instance, using the Sun: Gabella et al., 2016; Hubbert, 2017,) both Transmitter and Receiver chains, using a 24 GHz vertically pointing radar and disdrometers (Frech et al., 2017).

- Gabella, M.; Boscacci, M.; Sartori, M.; Germann, U. Calibration accuracy of the dual-polarization receivers of the C-band Swiss weather radar network. Atmosphere, 2016, 7, 76.
- Reimann, J.; Hagen, M. Antenna pattern measurements of weather radars using the Sun and a point source. J. Atmos. Ocean. Technol. 2016, 33, 891–898.
- Frech, M.; Hagen, M.; Mammen, T. Monitoring the absolute calibration of a polarimetric weather radar. J. Atmos. Ocean. Technol. 2017, 34, 599–615.

However, maybe, using a more robust definition for the SPR-GR reflectivity Bias, it will come out that the jump is smaller than 3.9 dB; for what concerns the assessment of the Mean Field Bias and its statistical evaluation, please have a look at following point #3)

**RESPONSE:** As of today, we were unable, despite repeated attempts, to retrieve detailed information on maintenance operations from the radar operator. The only information provided by PAGASA engineers was that hardware replacements happened in 2014 and 2015, and the supplier changed in 2016, as already elaborated in our response to Short Comment #1 (comment no 3). As much as we appreciate the recommendations and background information on transmitter and receiver chains provided by the referee, we hope he agrees that, based on the level of information provided by the operator, any discussion of specific causes for the jumps would remain speculatory. We agree with the referee that this lack of information - that should exist  somewhere - leaves us somehow dissatisfied.

**ACTION:** As already pointed out in our response to Short Comment #1 (comment no 3), we added a brief statement in at the end of 4.2(3):
*We have to assume that a fundamental issue with regard to calibration maintenance was addressed between 2013 and 2014 from hardware changes (i.e. replacement of magnetron). Unfortunately, we were not able to retrieve detailed information on maintenance operations that might explain the changes in bias of the radar throughout the years.*

**3)** From my viewpoint, the study is a bit limited in the definition of Bias assessment and the corresponding statistical metrics for the evaluation. For instance, it is going to be straightforward for the authors to derive other statistical parameters and present them in a summarizing Table that can complement the nice and informative figure 8. First of all, in addition to the annual mean of {ΔZdB*} (lines 20-22, page 15) also the standard deviation of {ΔZdB*}. Then, I would suggest a more robust definition for the Bias: instead of using dBZ, you use Z values in linear units: Z=10(dBZ/10). Then you derive a weighted average for the numerator (denominator) using linear Z of the GR (SPR). Finally, you compute 10 Log of such ratio (dB). This annual Log_of_the_MFB is more resilient than the Mean_of_the_Log presented in the paper. To avoid weighted-average or in a probability matching scheme, you may want to consider only bins with QBBF larger than say, 0.9 (or larger). After having done this selection, consider the difference (BIASxx) between different quantiles (probability matching): xx=50 (median), 75, 84, 90, 95, 99 percentiles. Maybe, BIASxx is not constant, rather it depends on the percentile? Finally, for these QBBF-selected bins, you may explore the value of the average bias E{ΔZdB} as a function of the intensity of the echo of the GR (using for instance intervals of 3 or 5 dBZ; obviously, you will have less and less samples for larger values of dBZGR). Does this Mean_of_the_Log Bias remain more or less constant? Or do you see a trend? (Maybe, SPR has residual attenuation for large reflectivity values?). Interesting, is not it?

**RESPONSE:** We would like to thank the referee for sharing these ideas. We will implement the suggestions as follows:
- We will add a visual representation of the standard deviation of the annual mean {ΔZdB*} in figure 8;
- We will recompute the bias estimates based on the referee's suggestion to first convert reflectivity to linear units before computing the weighted average;
- We will analyse the sensitivity of results in case we replace the weighted average by a simple quality threshold below which the samples will be discarded in the computation of calibration bias; however, we have the feeling that the paper is already very long, so we suggest to put the results of that analysis in a supplementary and only briefly refer to that in the main paper. Of course, using only partial beam blocking as a quality variable has very specific implications as to the effect of thresholding: any additional sample that exhibits a higher degree of partial beam blockage, and that we include in our computation of average reflectivity, will lead to a lower estimate of average ground radar reflectivity in the sample. Then again, reducing sample size through excessive filtering increases the standard deviation. That problem cannot really be resolved, but using the weighted average appears to us as the least arbitrary solution;
- Finally, we will add an analysis in which we investigate the dependency of our bias estimate on the intensity of the ground radar echo. Again, we suggest to present the results of that analysis in a supplementary, and only briefly discuss them in the main text.

**ACTION:**
Dashed lines are added in Figure 8a to represent the standard deviation of the annual mean.

After some deliberation, we would argue that processing the data in linear units would bias the results to higher reflectivities. Especially in the case where there is less confidence in SR measurements at high

reflectivity (due to attenuation), that is something we definitely would like to avoid. In addition, since it is the bias in dB that we are interested in, this is the unit that should be used for averaging. We also could not confirm the hypothesis that the computing the bias in linear units would lead to more robust bias estimates, and, as a consequence, to a lower variability of bias estimates between overpasses.

Altogether, we appreciate the suggestion of the referee to consider an alternative way to calculate the mean bias, yet we decided to keep the original approach of taking the mean bias as the average of the residuals between $dBZ_{GR}$ and $dBZ_{SR}$.

As for the probability matching, we have to admit that our above (blue) response was based on a misunderstanding of the referee's comment: originally, we thought the suggestion was to test the effect of discarding samples below a certain quality threshold instead of computing a weighted average. After understanding this mishap, we carried out the suggested analysis of a probability (or quantile) matching: We related the values of GR reflectivity at varying percentiles to the same SR reflectivity percentile, and plotted the difference of the two percentiles ($P_{i,GR} - P_{i,SR}$) as a function of the percentile (i) itself. The results for the different years is shown in Figure A.

[Figure]

Figure A. Difference of $Z_{GR}$ and $Z_{SR}$ at varying percentiles

In an ideal case, the difference should not vary depending on the percentile. For all years, we can observe an increase of the differences for very low percentiles and very high percentiles. That is consistent with the findings from Warren et al. (2018): for low percentiles, the effect could be a direct consequence of the low sensitivity of the SR. For high percentiles, the increase might be related to the undercorrection of attenuation in the SR beam. Yet, for 2012, 2013, and, to a lesser extent, 2015, the bias difference decreases over a broad range of percentiles, reaches a minimum between 60 and 80 %, and then increases again. Unfortunately, we have not found an adequate explanation for that behaviour, yet.

The final analysis required by the referee relates closely to the probability matching, since it investigates how the estimated bias (or the difference between GR and SR) depends on ground radar reflectivity (hence, instead of looking at percentiles, we look at specific reflectivity classes in intervals of 1 dBZ). Instead of showing the average behaviour over a full wet season, we decided to look at specific overpasses, in order to avoid averaging over different conditions and processes over time. Figure B shows the median GR-SR reflectivity difference as a function of GR reflectivity for each overpass for the 20 overpasses with the highest number of matched samples. GR reflectivity class intervals with less than 50 matched points are shaded with a lighter color. As a consequence of this procedure, only very few overpasses provide us with a complete picture. For those overpasses, though, which have a sufficient number of samples over a wide range of reflectivity classes, the results are quite incoherent: some overpasses exhibit a similar behaviour as mentioned above (for the probability matching), and as also discussed by Warren et al. (2018): for low and high GR reflectivities, $\Delta Z$ tends to increase with reflectivity, while for intermediate reflectivities, $\Delta Z$ is rather constant. Other overpasses show a more continuous increase of $\Delta Z$ with GR reflectivity.

It might well be worth following up on these analyses: if we could understand this behaviour better, there might be reason to limit the computation of calibration bias on intermediate reflectivities, or on overpasses in which $\Delta Z$ does not substantially depend on reflectivity. At this point, however, the analysis distracts, in our opinion, too much from the focus of the paper - the effect of quality weighting. Furthermore, our understanding of these results is yet limited. As a consequence, we decided that we will not put the corresponding results neither in the main manuscript nor in a supplementary, and we hope that the referee can agree.

[Figure]

Figure B. Median GR-SR reflectivity difference as a function of GR reflectivity for each overpass for the 20 overpasses with the highest number of matched samples. GR reflectivity class intervals with less than 50 matched points are shaded with a lighter color.

**4)** Finally, the last issue is related to literature: while several TRMM PR vs GR papers are listed, there is a lack of DPR-related studies and DPR technical literature. regarding the latter, I have suggested at the end (GPM related references), three papers published in 2014 and 2015. Regarding the former, I have listed our recent DPR-related studies in the complex terrain of Switzerland; I am confident the authors will be able to find additional GPM papers also in other parts of the world. Furthermore, is Cao et al. double citation (at page 6) correct? Does Morris and Schwaller (2009) exist? (line 7, page 1 citation) Regarding

GPM, please, do not forget to mention that your analysis neglect Ka-band observations (please briefly discuss the reason of such a choice).

**RESPONSE:** Indeed the references were incomplete, missing DPR technical literature. References will be added accordingly, including intercomparisons between ground radar and the GPM DPR.
We will remove one of the references to Cao et al. (2013) on page 6, and correct the actual reference (as already pointed out above). The Morris and Schwaller (2009) citation mistake will also be corrected.

We will add a brief note that GPM Ka-band observations have not been considered in the present study, reasons being Ka band being more prone to attenuation, and limited validity of the Rayleigh scattering hypothesis in a substantial portion of rainfall cases (see e.g. Baldini et al. 2012).

**ACTION:** The DPR technical literature and GPM papers have been added. The Morris and Schwaller (2009) citation mistake is corrected. The redundant citation for Cao et al. (2013) has been removed.

Furthermore, we added the following sentence to section 2.1:
"*Precipitation radar data were gathered from TRMM 2A23 and 2A25 version 7 products [...] and GPM 2AKu version 5A products [...].* **Ka band observations have not been considered due to higher susceptibility to attenuation, and a limited validity of Rayleigh scattering in a substantial portion of rainfall cases (Baldini et al. 2012).** *From the collection of overpasses within these dates [...]*"

[...]

**5)** Introduction
- Line 4: ... to monitor the bias of the gauge adjustment factor to be applied to precipitation estimates of the GR.
- Lines 6: ... to quantify the GR reflectivity bias with respect to the reference (namely, SR reflectivity value after conversion from Ku-band to S-band).

**RESPONSE:** The above mentioned lines in the abstract will be revised accordingly.
**ACTION:** Proposed changes have been implemented.

**6)** In fact, I would propose the following terminology:
- Setting the Bias as close as possible to 0 dB between radar QPE and in situ measurements: *gauge-adjustment*
- Assessing the Bias between reflectivity of two radars: *relative calibration*
- Forcing to 0 dB the Bias in measured Power (dBm) between an external or internal reference Noise Source and the radar at hand: *absolute calibration*

**RESPONSE:** We agree with the suggestion and will revise the manuscript accordingly, introducing the labels "gauge adjustment", "relative calibration", and "absolute calibration" in the first section.

**ACTION:** Proposed changes have been implemented.

**7)** Section 2.1.2

The fact that only 283 overpasses were within the selected, reasonable 120 km range should be mentioned here. There is no reason to wait until former(see **) sec. 3.1.1. Similarly, you can at least anticipate that the number will considerably decrease upon conditional requirements such as min. # of "wet" pixels, time difference, min. # of bins above both GR and SPR sensitivity.

**RESPONSE:** We agree that it might be confusing to state numbers of overpasses or valid samples without already anticipating the effect of spatial limitations or additional filter requirements. We will revise the manuscript accordingly by stating these effects early in the paper.

**ACTION:** The paragraph in section 2.1.2 is updated as follows:
*[...] from 1 June 2014 to 31 December 2016.* ***From the collection of overpasses within these dates, only 183 TRMM overpasses and 103 GPM passes were within the radar coverage.*** *The data were downloaded [...]*

The paragraph in the Results and Discussion: SR-GR Matching (former section 3.1.1.) has also been updated accordingly:
*From the 183 TRMM and 103 GPM overpasses that intersected with the 120 km Subic radar range, only 74 TRMM and 40 GPM overpasses were considered valid after applying the selection criteria listed in Table 2.*

**8)** Question: have you only used only months from June to November? Not clear from the text. Please rephrase. In fig. 8, I see two overpasses in December (2012 and 2014). By the way, in Dec. 2012 E{ΔZdB*} is almost 5! dB, while the annual average is -4.1 dB?!? (see my previous points 2) and 3) )

**RESPONSE:** We used only the months from June to December, which coincides with the rainy season in the area. We will clarify that in the text. As for the case of December 2012, upon checking the particular GR-SR match (December 5, 2012), the value of $\Delta Z^*$ is indeed very high (4.2 dB) compared to the average. Looking at the GR and SR data, the number of samples seems sufficient (n=382), and the GR overestimation is consistent for the different elevation angles. As a result, we cannot provide a consistent explanation for this outlier.

**ACTION:** We clarified the use of only the months of June to December in the text.

**9)** By the way, I would propose the following structure for Sections and Subsections
2. Data
2.1 Spaceborne Precipitation Radar (SPR)
2.2 GR

3. Method
3.1 Partial beam shielding and quality index based on beam blockage fraction
3.2 SPR-GR volume matching

3.3 Assessment of the average reflectivity Bias

Section 3.1: I would move it (including former fig. 4) inside the new Section 3.1 (former Sec. 2.2)

4. Results and Discussion
4.1 Single event comparison
4.1.1 Case1
4.1.2 Case2
4.2 Overall June-November comparison during the 5-year observation period

**RESPONSE:** We would like to thank the referee for this suggestion. We agree with the proposed structure, and will update the paper accordingly.

**ACTION:** The structure has been revised according to the referee's suggestion.

**10)** Page 5, Line 4-5: Please delete the sentence, the reader is able to read the simple algebra in eq. (1).

**RESPONSE:** The sentence will be deleted.

**ACTION:** The sentence has been deleted.

**11)** Page 10, Lines 19-25: misleading. I cannot possibly agree. On the contrary, my interpretation is that partial beam blockage plays approximately the same role (0.7 dB difference between the silly estimate that include blockage and the conservative one that exclude all cases where BBF > 0.5). Please rephrase.

**RESPONSE:** We agree that our interpretation was hard to follow, and the corresponding paragraph kind of confusing. That was also pointed out in comment #11 of referee #2. We will rephrase that part of the paper, and hope that our point will become clearer - because we still think it is quite an important one! At a low elevation angle, substantial parts of the sweep are affected by **total** beam blockage. The affected bins are either below the detection limit, or they do not exceed the GR threshold specified in Table 2 of the manuscript. As a consequence, these bins will not be considered in our matched samples, and will thus not influence our bias estimate - irrespective of using partial beam blockage as a quality filter. At a higher elevation angle, though, the same bins might not be affected by **total** beam blockage, but by **partial** beam blockage, as also becomes obvious from Fig. 4 of the manuscript. If we consider these bins in our matched samples, they will cause a systematic error in our estimate of calibration bias, unless we use the partial beam blockage fraction as a quality filter by computing a quality-weighted average of reflectivity. As a consequence, the effect of quality-weighted averaging (with partial beam blockage fraction as a quality variable) can be most pronounced at "intermediate" elevation angles, depending of course on the specific topography and the relative position of the ground radar. We had referred to that effect as "counterintuitive" since one might naively expect that the detrimental effects of beam blockage on our estimate of calibration bias would *generally* decrease with increasing elevation angle.

**ACTION:** The part in the paper is rephrased as follows:

*This case demonstrates how partial beam blockage affects the estimation of GR calibration bias. At a low elevation angle, substantial parts of the sweep are affected by **total** beam blockage. The affected bins are either below the detection limit, or they do not exceed the GR threshold specified in Table 2. As a consequence, these bins will not be considered in the matched samples, and will thus not influence the bias estimate - irrespective of using partial beam blockage as a quality filter. At a higher elevation angle, though, the same bins might not be affected by **total** beam blockage, but by **partial** beam blockage, as also becomes obvious from Figure 4. Considering these bins in the matched samples will cause a systematic error in the estimate of calibration bias, unless we use the partial beam blockage fraction as a quality filter by computing a quality-weighted average of reflectivity. As a consequence, the effect of quality-weighted averaging (with partial beam blockage fraction as a quality variable) can be most pronounced at "intermediate" elevation angles, depending on the specific topography and the relative position of the ground radar.*

**12)** Page 13: Would you please add a complementary figure at ELEV= 1.5 for the 1.10.2015 overpass? Just like you did for the 8.11.2013 overpass.

**RESPONSE:** We thank the referee for the suggestion, however, we are hesitant to add the additional figure as it does not provide additional insight as compared to the comparison of two sweeps for 2013-11-08, while adding to the length of the manuscript. As a compromise, we suggest to add the additional figure to the supplementary material.

**ACTION:** We decided not to include a supplementary section and thus also not the additional figure for the higher sweep for the 1.10.2015 overpass. While it visually reaffirms the discussion made with respect to the effect of higher elevation angles on the partial beam blockage, it does not provide an additional insight. We hope that the referee can agree.

**13)** Page 14 and Figure 8. Some journals ask for a graphical abstract as a self-explanatory image to appear alongside with the abstract. I think Fig. 8 would be perfect for such scope. It is nice and rich of information. Suggestion: could you please use color. For instance, the 1.10.2015 and 8.11.2013 overpasses could be in color. By the way, the 8.11 circle in picture a) seems to be very close to 0 dB, while in Fig. 7 it is written that $E\{\Delta ZdB^*\}$ is -1.1 dB. Am I missing something? Is it related to what you wrote in lines 3-6? These sentences are not clear to me, could you rephrase, please? Furthermore, regarding picture b), do not forget to emphasize that if the QBFF works properly then: $E\{\Delta ZdB^*\} - E\{\Delta ZdB\}$ should be negative in 2012 and 2013 (almost all the point in a) are below the 0 dB dotted line), positive in 2014 (almost all the point in a) above the 0 dB dotted line …).

**RESPONSE:** We thank the referee for the suggestion to highlight the two case studies in Figure 8 by color, and we will implement the suggestion accordingly. We are also grateful for suggesting a potential error, however, in this case, we do not agree: the triangle for Nov 8, 2013, represents correctly the bias

estimate on that date, as an average over samples from all sweeps (-3.7dB). Apart from that, Fig. 7 refers to the overpass on October 1, 2015.

We also thank the referee for pointing out the issue of negative differences E{ΔZ*} - E{ΔZ} in Fig. 8b which we missed to discuss sufficiently in the manuscript. First, we would like to clarify that if the QBBF works properly, the difference E{ΔZ*} - E{ΔZ} should be positive - the areas suffering from partial beam blockage registers weaker signals (i.e. lower reflectivity) than expected producing the "old" lower mean bias, and giving them low weights in the calculation of mean bias brings the "new" (quality-weighted) mean bias up. In the same vein, the difference in standard deviation should be negative - the "new" standard deviation that considers quality is lower than the "old" standard deviation that does not consider quality, so that the difference between "new" and "old" standard deviation is negative. The negative differences E{ΔZ*} - E{ΔZ} are therefore inconsistencies, caused by the effect of filtering in the case of very small sample sizes. We will include this clarification in the revision.

**ACTION:** In Figure 8, we have added green vertical lines at the dates of the case studies so as not to interfere with the color symbolism of the markers and bars.

The paragraph in Section 4.2(1) on **Effect of quality weighting on bias estimation** has been updated to:
*Figure 8b and c together illustrate the benefit of taking into account GR data quality (i.e. beam blockage) when we estimate GR calibration bias. It does not come as a surprise that the difference between ΔZ\* and ΔZ is mostly positive because the areas suffering from partial beam blockage register weaker signals (i.e. lower reflectivity) than expected, producing a lower mean bias. Giving the associated volume-matched samples low weights in the calculation of the mean bias brings the quality-weighted bias up. In the same vein, the beam-blocked bins introduce scatter, and assigning them low weights decreases the standard deviation. Figure 8c shows, as a consequence, that the quality weighted bias estimates are consistently more precise: in the vast majority of overpasses, the quality weighted standard deviation is substantially smaller than the simple standard deviation. That result is also consistent with the case study result shown above. It should be noted, though, that for some overpasses, the quality weighting procedure (which is in effect a filtering) can cause an increase in the bias estimate and/or the standard deviation of that estimate. That effect occurs for overpasses with particularly low numbers of matched samples, and, presumably, with rainfall in regions in which our estimated beam blockage fraction is subject to higher errors (caused by e.g. the inadequateness of the assumed Gaussian antenna pattern, variability of atmospheric refractivity, or errors related to the DEM, its resolution and its interpolation to ground radar bins). In total, however, the effect of decreasing standard deviation vastly dominates.*

**14)** Page 15. I would change the order of your points and list your point (1) at the end, as # (4) [see my comment 1) at page 1)]. I would start from (3), which is the scope of this paper: indeed an intelligent weighted-average based on QBFF shows a better standard deviation of ΔZdB*. By the way, I recommend adding a table and/or a figure (histogram) that summarizes the statistical properties of $\sigma$*{ ΔZdB*} and $\sigma${ΔZdB}. Then, I would introduce the important result regarding the consistency of GPM and TRMM radars, followed by the changes of the bias in time

**RESPONSE:** As already pointed out in our response to comment #1 of the referee, we will change the order of points as suggested. However, we decided not to introduce additional figures in terms of histograms of bias, differences in bias, or standard deviations. These histograms would have to be provided separately for each year, because it is obvious from the time series that they would represent different populations. Apart from avoiding to introduce many new figures, the informative value of these histograms is not too high due to the limited number of samples. Instead, we will implement the referee's suggestion from comment #3 by including the standard deviation of the annual mean {ΔZdB*} in Fig. 8a.

**ACTION:** The order of discussed points has been changed (see also our ACTION to comment 1 of this referee). We modified FIg. 8 by showing the standard deviation of the mean annual bias by dashed lines. Please note that mean and standard deviation of the annual bias are not computed from the mean bias of each overpass, but from the total set of matched samples of an entire wet season. That is why the standard deviation is wider than the actual variability of the bias between overpasses.

**15)** Page 16.
Line 5, delete coherent.
**RESPONSE:** The word "coherent" will be deleted.
**ACTION:** The word "coherent" has been deleted.

**16)** Line 14-16. Sorry, you cannot summarize the (mis-) calibration of the GR by simply going from 2012 (–4.1 dB) to 2016 (+0.6 dB) and omit, for instance, the +1.4 jump in 2014. [see my comment
2) at page 2)].

**RESPONSE:** We will revise the manuscript accordingly by providing a more complete and coherent summary of the temporal changes of our bias estimate.

**ACTION:** The sentence has been revised to provide a more complete summary of the temporal changes of the bias estimates, and now reads as:
*Analyzing five years of archived data from the Subic S-band radar (2012-2016), we also demonstrated that the calibration standard of the Subic radar substantially improved over the years, from bias levels around -4.1 dB in 2012 to bias levels of around 1.4 dB in 2014 and settling down to a bias of 0.6 dB in 2016. Of course, further studies looking at more recent years are necessary to evaluate the steadiness of the bias.*

**17)** Line 17-19. Pleonastic. I would delete it.

**RESPONSE:** We would like to refer to our response to the referee's comment #11: we hope that we were able to clarify a misunderstanding there. Given that the referee agrees with our clarification, we think that lines 17-19 on page 16 are not pleonastic, but rather an important note to emphasize that moving to higher elevation angles does not necessarily help to avoid the problems introduced by beam

blockage in the specific case of comparing GR and SR observations. Nevertheless, we will also revise the corresponding paragraph in the conclusions section in order to make it more comprehensible.

**ACTION:** We hope that the revision of the paragraph in the discussion of Case Study #1 has clarified the misunderstanding. We wanted to emphasize that moving to higher elevation angles does not necessarily help to avoid the problems introduced by beam blockage in the specific case of comparing GR and SR observations. With the clarification, we think that the corresponding lines in the conclusion is now comprehensible as it is.

**18)** Line 26. Why do you discuss C-band radar technology ?

**RESPONSE:** Lines 24-28 on page 16 of the original manuscript were intended to provide a brief perspective for future studies, in which we mention that for C-band radars, it would be important to include path-integrated attenuation as a quality variable. In the revised version, we will clarify that point.

**ACTION**: In the conclusion (section 5), we changed the corresponding sentence as follows:
*For example, if we consider C-band instead of S-band radars, path-integrated attenuation needs to be taken into account for the ground radar, and wet radome attenuation probably as well (Austin, 1987; Merceret, 2000; Villarini and Krajewski, 2010).*

**19)** Minor points
My proposal for radar acronyms: 2-character for ground, namely GR; 3-character for satellite radar. Would you please use TPR for TRMM, DPR for GPM and SPR in those cases where you refer to both, independently of the platform

**RESPONSE:** We appreciate the suggestion. Yet, we think that distinguishing the different spaceborne platforms via acronyms might cause more confusion than clarification, in particular since we rarely address the different platforms separately in the main text. We would thus prefer to stick with GR vs. SR in general.

**ACTION:** Based on our above (blue) response, we decided to keep the acronyms as is.

**Referee Comment #2 (Anonymous)**

[...]

**Comments on other sources of uncertainty in calibration assessment:**

**1)** Attenuation at Ku-band:
The authors should address the uncertainties with attenuation correction at Ku-band. The

attenuation correction tech. used for just Ku-band is the HB-SRT method (Seto and Iguchi 2015). It is known that using the HB method alone does not work well in higher rain rates (> 20 mm hr-1, Seto and Iguci 2011, but as low as 12 mm hr-1 Rose and Chandrasekar 2005). Furthermore, the SRT method is more uncertain over land (larger standard deviation of the surface backscatter cross-section, Meneghini et al. 2000). It is anticipated that since the radar is located in the tropics both of the issues above could occur (more likely in convective precipitation). Please discuss these uncertainties and how they could impact your results of the bias correction.
It is mentioned in the conclusions that for C-band attenuation correction is vital, but GPM and TRMM are Ku-band, thus isn't it vital as well?

**RESPONSE:** We agree that attenuation correction is vital for both GPM and TRMM at Ku-band, and there is certainly a large body of literature concerned with the related effects, including the effects of nonuniform beam filling (NUBF) on the attenuation correction procedure. In the present study, we have only used the attenuation-corrected reflectivity values without considering the uncertainty associated with the correction procedure. In the revised manuscript, we will explicitly refer to the uncertainty introduced by attenuation correction. We will also, in the conclusions, provide an outlook on including the spaceborne reflectivity observations in the framework of quality-weighted averaging, just as we suggested for the ground radar observations. That would imply to use the estimates of PIA which are provided through the SR meta-data as a quality variable and thus to consider it in the quality-weighted average of SR reflectivity in the matched samples.

**ACTION:** We have addressed attenuation (correction) at Ku band explicitly in our discussion of uncertainties in section 4.2(4).  In the conclusions (section 5), we provided an outlook on including SR observations in the framework of quality-weighted averaging, with the addition of this sentence:
*The framework could also be extended by explicitly assigning a quality index to SR observations, too. In the context of this study, that was implicitly implemented by filtering the SR data e.g. based on bright band membership. An alternative approach to filtering could be weighting the samples based on their proximity to the bright band, the level of path-integrated attenuation (as e.g. indicated by the GPM 2AKu variables **pathAtten** and the associated reliability flag (**reliabFlag**)) or the prominence of non-uniform beam filling (which could e.g. be estimated based on the variability of GR reflectivity within the SR footprint, see e.g. (Han et al., 2018)).*

**2)** Ground Clutter for the SR:
In radar gates near the surface, with respect to the SR, ground clutter is a problem. How are the authors dealing with ground clutter from the SR? Are they using gates below the lowest clutter free bin estimate (included in the GPM file)? If so, is the lowest clutter free gate being assigned to all the gate below it? If you plot it out, a lot of times that's what is done. Essentially the data looks smeared from the lowest clutter free bin to the surface, which isn't to realistic and it is suggested to just not consider these gates. Please comment on this, potentially in Section 2.3. If you are including these interpolations, you may wish to not (it will introduce error).

**RESPONSE:** Thanks for pointing out this issue which has not yet been sufficiently clarified in the original manuscript. While TRMM 2A25 contains a clutter flag for the variable "Corrected Z-factor" (-8888 indicates ground clutter), the GPM 2AKu product contains a variable "binClutterFreeBottom" to indicate the lowest clutter free bin in a ray. In both cases, TRMM and GPM, we use the SR clutter information to discard the affected bins. We will clarify that point in the revised manuscript, using both table 2 (filtering criteria), and the new table with metadata variables that we introduced as a response to comment #7 of SC1 (as part of the the supplementary).

**ACTION:** First, we would like to apologize for a mistake we made in our initial response. We had in fact not considered the lowest clutter free bin for the GPM data in the submitted manuscript. We discarded clutter affected bins based on TRMM clutter flags, but did not explicitly consider the variable "binClutterFreeBottom" for GPM. We would like to thank the referee for bringing up this point. Upon investigating the binClutterFreeBottom variable (i.e. the lowest clutter-free bin), we have noticed that the average altitude of that bin is at about 3000 meters (with a range of 1300m-7200m) in the study area. Given that the highest mountain peaks are around 2000 meters, we do not have much confidence in that variable. Assuming clutter-contaminated SR bins at such altitudes and thus discarding the corresponding SR bins would exclude the majority of the volume covered by our ground radar. We thus decided to use the GPM data as is, although the poor clutter identification surely merits further attention in the future, and has also been confirmed by other studies (e.g. Watters et al., ERAD 2018 book of abstracts)

**3)** NUBF:
Please also include some discussion of the potential impacts of non-uniform beam filling (NUBF) on your analysis. Edges of large systems, individual cumulus showers could result in NUBF in SR because of the quasi-large footprint. Lowering the reflectivity value in the gate.

**RESPONSE:** We agree that non-uniform beam filling can cause errors in particular for the SR platform which might become more pronounced in case of path-integrated attenuation is present and being corrected for. Durden et al. (1998) provided an excellent discussion of potential effects. Han et al. (2018) attempted to consider the effect in case GR and SR observations are matched, by using the - comparatively highly resolved - GR observations in order to compute the standard deviation of reflectivity in an SR footprint as a measure of NUBF. From the literature, it is hard to tell how much systematic error is introduced in SR measurements by the effects of NUBF. However, the three comments of this referee (reg. attenuation, clutter, NUBF) were very helpful for us to understand the necessity of extending the framework of quality-weighted averaging to the SR, too. So while we consider our present manuscript as a proof-of-concept in the consideration of quality, follow up studies should attempt to achieve a more general implementation that not only includes additional quality variables for the GR data, but that also applies these to the SR observation which already come with extremely rich and helpful meta-data to support such attempts. While our study tries to minimize the effects of NUBF (by setting a minimum fraction of GR bins within the SR footprint to exceed a minimum reflectivity threshold, see table 2 of the original manuscript), a future framework for SR quality might rather consider the variability of GR bins in the SR footprint, as suggested by Han et al. (2018).

**ACTION:** We explicitly included NUBF in our list of uncertainties in section 4.2(4). Furthermore, the following sentence was added to the conclusions (section 5):
*[...] proximity to the bright band,* **the level of path-integrated attenuation (as e.g. indicated by the GPM 2AKu variables "pathAtten" and the associated reliability flag ("reliabFlag"), ) or the prominence of non-uniform beam filling (which could e.g. be estimated based on the variability of GR reflectivity within the SR footprint, see e.g. Han et al. (2018))***. In addition, [...]"*

**Specific Comments:**

**4)** Page 2, line 5: Please add the Kummerow et al. (1998) paper for TRMM, and the Hou et al. (2014) for the GPM reference (page 2, line 6). This will help readers who are not entirely familiar with both platforms.

**RESPONSE:** The Kummerow et al. (1998) and Hou et al. (2014) citations and references will be added.
**ACTION:** The references have been added.

**5)** Page 6, line 3: "The gates below and above the brightband were considered in the comparison". Please provide a brief reason why this is done. I do not want to assume the author's reasoning.

**RESPONSE:** According to Warren et al. (2018), the frequency-corrected reflectivities within the melting layer (bright band) appear underestimated compared to the ones below and above the melting layer. In addition, while usually the samples above the brightband are used in GPM validation, there are significantly more samples below the melting layer, especially in a tropical environment such as the Philippines.

**ACTION:** The following explanation is added in section 3.2:
*...Only gates below and above the brightband were considered in the comparison. Warren et al. (2018) found a positive bias in GR-SR reflectivity difference for volume-matched samples within the melting layer, compared to those above and below the melting layer. They speculated that this was due to underestimation of the Ku- to S-band frequency correction for melting snow. In addition, while usually the samples above the brightband are used in GPM validation, there are significantly more samples below the melting layer, especially in a tropical environment such as the Philippines. To ensure that there are sufficient bins...*

**6)** Figure 4 & Section 3.1: It is not clear what you are plotting. The figure titles state the quality index but the figure caption and text states beam blockage fraction. Please clarify.

**RESPONSE:** The caption has been updated to match the figures: *Quality index map of the beam blockage fraction for the Subic radar at (a) 0.0° (b) 0.5° (c) 1.0° and (d) 1.5° elevation angles.*

**ACTION:** The caption has been updated.

**7)** Section 3.1.1: Why are the number of overpasses here different than when they were listed earlier (section 2.1.2)? I am referring to the numbers before applying the criteria in Table 2.

**RESPONSE:** Applying the criterion of "Minimum number of pixels tagged as rain = 100" eliminates several overpasses. Only this criteria affects the number of overpasses, not the others listed in Table 2. We will clarify this in the paper.

**ACTION:** The number of overpasses stated in section 2.1 has been updated to reflect the number of overpasses that intersected with the radar coverage. This is now consistent with the numbers mentioned in the results section.

**8)** Case studies (Section 3.1.2 and 3.1.3): Could you include the mean BB level height? You can add it to the bottom right with the other statistics. Also comment on fraction of stratiform vs convective. These two will help readers assess the amount of attenuation and NUBF that could be involved (e.g. uncertainty in the SR measurements).

**RESPONSE:** The mean BB level height will be added to the figure as suggested. While stratiform rain dominates the precipitation type for most cases, convective rain is significantly represented, hence we decided to keep both rain types in the analysis.

**ACTION:** We added the mean BB level height to the caption of Figures 5 and 7.
*Figure 5. [...] The mean bright band level is at a height of 4685 meters.*
*Figure 7. [...] The mean bright band level is found at 4719 meters for this case.*

**9)** Figure 5 + 6 + 7 a and b: Suggestion. Consider changing the colorscale to one that is perceptually uniform and color-deficient friendly. For example, try the HomeyerRainbow or the LangRainbow included in Pyart (https://github.com/ARM-DOE/pyart)

**RESPONSE:** We thank the referee for the suggestion. Upon trying the different colormaps proposed, we decided that we will go with the HomeyerRainbow colormap. The figures will be updated to reflect the new colormap.

**ACTION:** The colormaps for Figure 5+6+7 a and b have been updated to follow the HomeyerRainbow colormap in PyART. The scale of the colormap for subplots c and d have also been shifted such that the lowest value is darker than the previous version, for better visibility, following the suggestion in comment (9) of Short Comment #1. The point size of the scatter plot has also been reduced to minimize overlaps.

**10)** Page 10, Line 12: "Major parts of that sector did not receive any signal due to total beam blockage". Where is this occurring? The reader can refer back to Figure 4, but it might be helpful to outline the circles with a thin black line in Figure 5d where there is SR data,

but no GR data. That way the readers would see where there is 100% beam blockage and thus no signal from the GR, but also gain insight of size of the precipitating system.

RESPONSE: The figures for the case studies show only the matched bins, but the referee is right, information such as location of bins where there is SR signal but no GR signal and the size of precipitating system are not conveyed. We will address this by showing all the available SR bins for the first panel and outlining the circles with SR data but no GR data in black, as suggested.

ACTION: In Figures 5-7, the SR bins where SR data is present but not GR data is encircled in black, as suggested. Correspondingly, the text in Section 4.1.1 now reads:
*Major parts of that sector did not receive any signal due to total beam blockage**, highlighted in Figure 5a with black circles showing the bins where the GR did not have valid observations.***

**11)** Page 10, Line 24-25: "That might be considered counterintuitive, as one might expect the blockage to disappear with higher elevations". Please provide some discussion explaining why this is the case.

RESPONSE: We thank the referee for pointing out the lack of adequate explanation. As can be seen also from the comments of referee #1, this paragraph appears to be confusing in the original manuscript. We will revise the paragraph accordingly in order to make our point clearer. Please also refer to our response to the comment #11 of referee #1.

ACTION: The confusing paragraph has been updated and a better discussion of the effect of "intermediate" elevation angles on the partial beam blocking has been included. Please also refer to our response to the comment #11 of referee #1.

**12)** Page 16, Lines 13 – 16. 'We could' and 'we could also' imply that you did not conduct this analysis when it seems you have. I suggest to change these phrases to be definitive. 'We showed that…' 'we also demonstrated that…'
RESPONSE: The sentences will be updated as suggested.
ACTION: The sentences have been updated as suggested.

**Technical corrections:**

**13)** Page 3, line 20: The most current GPM version is version 5, version 6 is not released yet.
RESPONSE: The version will be corrected (version 5A instead of 6).
ACTION: The version has been corrected (version 5A instead of 6).

**14)** Page 18, line 18: Reference Cao et al. 2013 is incorrect. It should be:
Empirical conversion of the vertical profile of reflectivity from Ku-band to S-band frequency

RESPONSE: We apologize for the mixup. The citation and reference will be corrected to refer to

Cao, Qing, Yang Hong, Youcun Qi, Yixin Wen, Jian Zhang, Jonathan J. Gourley, and Liang Liao. 2013. "Empirical Conversion of the Vertical Profile of Reflectivity from Ku-Band to S-Band Frequency." *Journal of Geophysical Research: Atmospheres* 118 (4): 1814–25. https://doi.org/10.1002/jgrd.50138.

**ACTION:** The citation and reference has been corrected.

**15)** The reference Warren et al. should be 2018, published Feb 2018 in J. Atmo. + Ocean. Tech.. Page 2, line 8;Page 3,line 25;Page 5,line 11;Page 15,line 14
**RESPONSE:** The citations and reference will be corrected.
**ACTION:** The citations and references have been corrected.

**16)** Figure 4: Missing y-ticks and tick labels on bottom left subplot
**RESPONSE:** Axis labels will be restored in Figure 4. The color scheme has been changed so that the lightest color is made a bit darker for better visibility in Figures 5-7 subplots d and e, following the suggestion of another reviewer.
**ACTION:** The axis labels have been restored.

**17)** Page 8, line 5-6. No need for new paragraph. You can combine the two.
**RESPONSE:** The paragraphs will be combined as suggested.
**ACTION:** The paragraphs have been combined.

**18)** Figure 5: Figure caption has Zpr instead of Zsr
**RESPONSE:** $Z_{PR}$ will be replaced with $Z_{SR}$ in the caption
**ACTION:** The caption has been updated.

[revised manuscript text omitted]